# Wireless battery free fully implantable multimodal recording and neuromodulation tools for songbirds

Jokubas Ausra [1], Stephanie J. Munger[2], Amirhossein Azami[1], Alex Burton[1], Roberto Peralta [3],
Julie E. Miller [2,4✉] & Philipp Gutruf [1,5✉]

Wireless battery free and fully implantable tools for the interrogation of the central and peripheral nervous system have quantitatively expanded the capabilities to study mechanistic and circuit level behavior in freely moving rodents. The light weight and small footprint of such devices enables full subdermal implantation that results in the capability to perform studies with minimal impact on subject behavior and yields broad application in a range of experimental paradigms. While these advantages have been successfully proven in rodents that move predominantly in 2D, the full potential of a wireless and battery free device can be harnessed with flying species, where interrogation with tethered devices is very difficult or impossible. Here we report on a wireless, battery free and multimodal platform that enables optogenetic stimulation and physiological temperature recording in a highly miniaturized form factor for use in songbirds. The systems are enabled by behavior guided primary antenna design and advanced energy management to ensure stable optogenetic stimulation and thermography throughout 3D experimental arenas. Collectively, these design approaches quantitatively expand the use of wireless subdermally implantable neuromodulation and sensing tools to species previously excluded from in vivo real time experiments.

---

[1] Departments of Biomedical Engineering, The University of Arizona, Tucson, AZ, USA. [2] Department of Neuroscience, The University of Arizona, Tucson, AZ, USA. [3] Department of Aerospace and Mechanical Engineering, The University of Arizona, Tucson, AZ, USA. [4] Departments of Speech, Language & Hearing Sciences, Neurology, and Bio5 Institute, Neuroscience GIDP, The University of Arizona, Tucson, AZ, USA. [5] Departments of Electrical and Computer Engineering, Bio5 Institute, Neuroscience GIDP, The University of Arizona, Tucson, AZ, USA. ✉email: juliemiller@email.arizona.edu; pgutruf@email.arizona.edu

Recent advances in resonant magnetic power transfer[1], highly miniaturized device footprints[2], and digital control of wireless and battery free and fully implantable systems[3] for the modulation of the central[4] and peripheral nervous system[5] result in highly capable platforms that enable stimulation and recording of cell type specific activity in the brain and the peripherals. The systems quantitatively expand the experimental paradigms that can be realized with rodent subjects through the ability to stimulate and record with multiple animals simultaneously[6] and at the same time have minimal impact on the subject behavior[2] enabling recordings in naturalistic ethologically relevant environments.

The use of such devices has been almost exclusively demonstrated in rodents due to the availability of the genetic toolset for optogenetics[3,7] and genetically targeted fluorescent indicators[6]. The capabilities of these platforms however enable in principle the modulation of the central and peripheral nervous system beyond common animal models. Flying subjects have previously been only rarely used for optogenetic modulation[8] as they are commonly ruled out as species to perform behavioral experiments with due to the lack of optogenetic tools that can support their free movement. Freely moving songbirds represent a particularly interesting species to target with optogenetic devices due to their extensive use as model organisms to study mechanisms for human vocal learning and production[9]. The development of optogenetic tools in songbirds offers the ability to finely tune neural activity in specific pathways in real-time and affect vocal performance[10,11]. However, as shown in rodent species, the current methods of stimulation via optical fibers limit animal mobility[6], induce stress[12], cause mechanical damage[13], and require advanced cable management[14]. These limitations essentially eliminate free flight and multi animal experiments in flying animals.

Wireless and battery free devices with the ability to optogenetically stimulate and record temperature have not been utilized in the context of flying species due to challenges in primary antenna design, device miniaturization, and digital data communication throughout the typically larger experimental arena volumes required for such subjects. Here we introduce two new concepts that enable the operation of wireless battery free devices in songbirds. We employ the use of deep neural network analysis of behavior to identify the volume most occupied by the subject and design the primary antenna to elevate power delivery in these regions as well as introduce advanced power management that tailors digital communication to energy availability on the implant to enable reliable data uplink throughout the experimental arena.

## Results

### Wireless, battery free multimodal neuromodulation devices for songbirds.
To achieve device properties that permit the use of wireless and battery free implantable devices in songbirds, device form factor, footprint, and size have to be tailored to the available subdermal space on the bird skull. In this work, we use zebra finches which belong to the family of songbirds, which have an average available head area of $0.825 \, cm^2$ (Supplementary Information Fig. S1) and typically weigh $14.4 \, g$[15]. This is a 10.5% reduction in available head space as compared to mouse animal models[16]. The resulting shape and form factor that allows for the highest energy harvesting capability and space to position optogenetic probes is an oblong outline that maximizes antenna size and conforms readily onto the skull of the subject (Supplementary Information Fig. S2).

The layered makeup of the device is comprised of two copper layers insulated by a layer of polyimide carrier, populated with surface mounted components on the device top and bottom side, and encapsulated with parylene and elastomeric materials for modulus matching (Fig. 1a).

The device platform is demonstrated with two device versions enabling stimulation and recording. Specifically, we created a device with capabilities in bilateral optogenetic stimulation (Fig. 1b) and a device architecture that enables both thermography and optogenetic stimulation (Fig. 1c). For either embodiment, the device relies on resonant magnetic coupling at $13.56 \, MHz$[1] which can cast energy through a primary antenna, utilizing a commercial power amplifier readily available at this operational frequency, to the secondary antenna located on the implant (Fig. 1d). The captured magnetic field yields a current that is rectified and regulated by a low dropout regulator (LDO) to drive digital and analog electronics that handle optogenetic stimulus, digitalization and wireless communication.

The bilateral optogenetic stimulation device has minimal electronic hardware requirements which results in a smaller footprint that enables facile placement of the stimulating probes that rely on blue stimulation micro-inorganic light emitting diodes (μ-ILEDs) controlled by the microcontroller (μC) with current limiting resistors R1 and R2. For the multimodal optogenetic stimulator and thermography device, an operational amplifier is used to condition signals from a ultra-small outline negative temperature coefficient (NTC) sensor to yield mK sensor resolution, a set of capacitors to store energy temporarily, and an infrared data uplink to relay signals wirelessly[2]. A photographic image of the backside of a device with thermography capabilities is shown in Fig. 1e, revealing the NTC sensor that can be placed in intimate contact with the skull to capture high fidelity thermal signatures of the subject. The inset of Fig. 1e shows an example of the device's temperature recording modality over a 4 h period.

A typical experimental arena including exposure and recording chamber is shown in a photographic image in Fig. 1f where antenna and tuning system are installed at the righthand chamber. An image of the finch perching with the infrared (IR) LED activated during a data transmission event is shown in Fig. 1g, highlighting the miniaturized form factor of the device enabling naturalistic motion and no protruding features that allow for cohabitation with other birds in the colony.

### Behavior guided antenna engineering.
In comparison to wireless subdermally implantable and battery free devices for the use in rodents, birds often occupy larger volumes of space. By comparison, standard mouse animal model experimental cage volumes are 45% smaller when compared to standard experimental arenas used for songbirds[1,7,17]. It is therefore important to maximize transfer efficiency in the most occupied locations of the experimental arena to enable power hungry applications that feature multimodal stimulation and recording. Because behavior of the animals can vary depending on experimental paradigm and species, we introduce a strategy that utilizes state of the art deep neural network analysis of subject behavior (DeepLabCut[18]), originally developed for rodents. A simple webcam recording can be sufficient to track the animal's movement and subsequently build spatial position analysis that can be used to adjust primary antenna parameters as outlined in the workflow in Fig. 2a.

Spatial position heat maps can be created for the top (Fig. 2b) and side (Fig. 2c) views used to record the animals. Details on deep neural network analysis can be found in the Methods section and in Supplementary Information Figs. S3, 4. The behavioral pattern indicates that most of the time is spent in the upper and lower half of the arena with close proximity to the arena walls. This behavioral pattern is maintained throughout the day as the bird frequently flies to the floor of the arena to get food and then

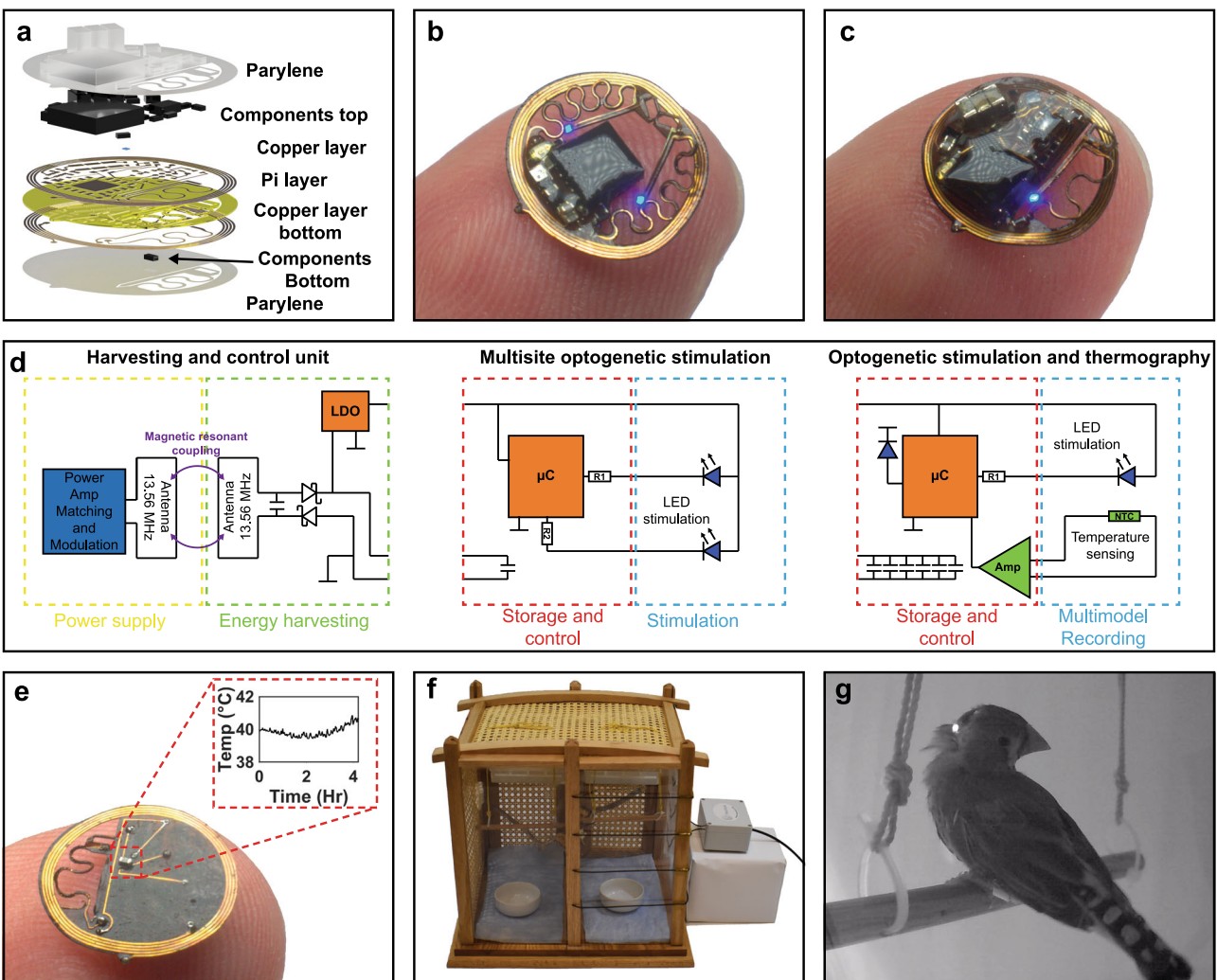

**Fig. 1 Device overview and basic operating principles. a** Exploded-view layout of the constituent layers of the system. **b** Photographic image of the dual optogenetic stimulation probe device balanced on a finger. **c** Photographic image of the multimodal optogenetic stimulator and thermography device balanced on a finger. **d** Schematic diagram of electrical operating principles of the two devices with harvesting and control unit (left), dual optogenetic probe stimulation circuit diagram (middle), and multimodal optogenetic stimulation and thermography circuit diagram (right). **e** Photographic image of the back of the multimodal optogenetic stimulator and thermography device with inset showing temperature sensing capabilities over a 4-hour period. **f** Photographic image of the experimental arena with stimulation arena in the right chamber. **g** Photographic image of finch perched in the experimental arena with IR LED activated during data uplink event.

returns to the perch located near the top of the arena. This spatial occupancy pattern however does not match the conventional antenna design introduced for wireless and battery free implants for rodents[7]. This primary antenna layout for a 20 cm x 35 cm x 35 cm experimental arena with equidistant antenna loops is shown in Fig. 2d and produces a relatively uniform power distribution with highest power values reached in the corners of the arena (23.34 mW) and average power at center of the arena volume (14.61 mW) found at a height of 9 cm from the cage floor (detailed measurement procedure outlined in the Methods section). This distribution however is not ideal for the behavioral patterns of the birds. To adjust the power distribution to match subject behavior, the primary antenna design was chosen in a Helmholtz-like configuration for the same arena size (Fig. 2e). This antenna arrangement produces a stronger resonant magnetic field in the lower and upper halves of the arena, with a power of 18.90 mW measured at the center for a height of 5 cm from the cage floor and a power of 16.64 mW at the center for a height of 25 cm, substantially increasing the power transfer at the horizontal levels most occupied by the birds. Maximum power, which is

typically achieved in the corners of the cage because of inductive coupling dominated regimes closer to the antenna, increases by 80% for the Helmholtz-like coil configuration (42.03 mW) compared to the standard configuration (23.34 mW). This approach to modify antenna geometry to tailor power delivery to the subject based on spatial position is ubiquitous, enabling new experimental paradigms. Strategies to preferentially deliver power are demonstrated in Supplementary Information Fig. S5. It is also possible to change arena dimensions significantly if the same volume is maintained. This is demonstrated in Supplementary Information Fig. S6a where an arena length of 105 cm with cross-sectional dimensions of 15 cm x 15 cm can sustain device operation. Arenas with increased volumes are also possible by combining multiple RF power sources to achieve coverage for larger volumes (Supplementary Information Fig. S6b–d) with arbitrary shape, significantly expanding the utility of the devices to investigate behavior with simultaneous neuromodulation.

Equally important for the optimization of power transfer is the antenna design of the secondary antenna on the implant. Antenna geometry is designed to occupy the largest possible

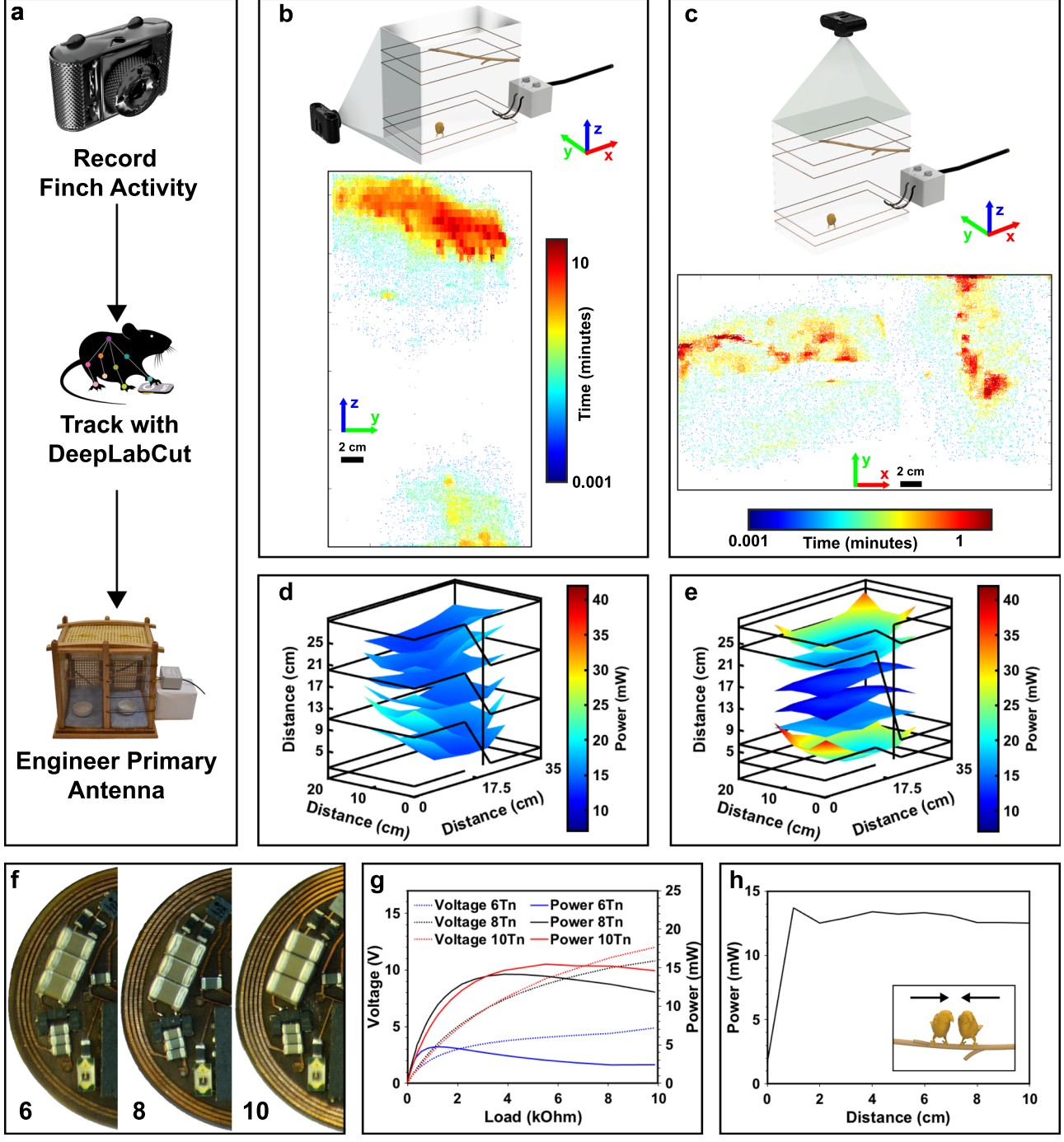

**Fig. 2 Behavior guided primary antenna design and secondary antenna optimization. a** Workflow diagram showing behavior guided antenna design for resonant magnetic coupling-based devices. **b** Rendered schematic of side camera recording setup (top) and heat map indicating spatial location of the finch. **c** Rendering of cage with top camera (top) and heat map indicating spatial location. **d** Spatial mapping of power distribution for the subdermally implantable device in an arena with dimensions 20 cm x 35 cm x 35 cm with 4 turn antenna equidistantly spaced and 10 W RF input power. **e** Spatial mapping of power distribution for the subdermally implantable device in an arena with dimensions 20 cm × 35 cm × 35 cm with 4 turn antenna spaced in a Helmholtz-like configuration with 10 W RF input power. **f** Photographic image of secondary antenna with 6, 8, and 10 turn configurations. **g** Power vs. load curve for the miniatured device at the center of the arena 25 cm from the floor measured with primary antenna in Helmholtz-like configuration and an input power of 10 W. **h** Available device power vs. distance between two wireless devices at the center of the arena 25 cm from the floor with wires in a Helmholtz-like configuration and an input power of 10 W.

area on the skull of the bird and maintain facile implantation. The secondary antenna is optimized by empirically comparing antenna configurations in the experimental arena (Fig. 2f). Antenna performance is compared at 5.6 V, which is the maximum operation voltage for the LDO and capacitors used

in this design. Typically, a higher number of turns result in a gain in harvesting ability because of a larger inductance of the antenna[19]. However, with higher inductance the maximum harvesting capability usually occurs at higher voltages which are not practical for use in highly miniaturized systems because small

outline components generally also feature lower operation voltages. Figure 2g shows that ultimate peak harvesting capability, tested in a 20 cm x 35 cm x 35 cm experimental arena at a height of 25 cm with 10 W RF input, was achieved by a 10-turn dual layer antenna (5 turns on the top and bottom layer of the device) with trace width of 100 μm and trace gap width of 50 μm which harvests a peak power of 15.46 mW at a load of 5.6 kΩ and 9.23 V. The 8 turn antenna, which has 4 turns on the top and bottom layers of the device with identical trace spacing as the 10 turn device, in contrast has a lower peak harvesting capability of 14.14 mW which occurs at 7.38 V. This antenna option outperforms the 10 turn variant at 6.0 V and below significantly, surpassing the 10 turn device by 1.35 mW at the typical operation voltage of the rectifier at high loads (3.5 V). This optimization enables higher harvested power and improves usable footprint for electronics, sensors, and neuromodulation probes. Harvesting capability of the antennas scale linearly with RF field input power and a characterization is shown in Supplementary Information Fig. S7. Angular harvesting capability, an important consideration in freely behaving subjects due to behaviors such as feeding that can decrease harvested energy, exhibits linear decrease of harvested power (Supplementary Information Fig. S8) which is consistent with behavior observed in previous work[1].

Birds are usually social animals that often cohabitate small spaces, because of this behavior an investigation into detuning of the secondary antenna for two subjects in close proximity is conducted. Results shown in Fig. 2h reveal that power harvesting capability is only affected if the devices are overlapping, which suggests stable operation of many devices even in close proximity of subjects.

**Stimulation control and characterization**. The small space available on the finch skull necessitates a significant reduction of electronics footprint by 37.5% compared to previous work in rodents[1]. At the same time, control capability has to increase due to requirements to stimulate many subjects simultaneously. To address these needs, a communication strategy leverages one wire-like communication protocols that utilize the EEPROM of the μC together with ON/OFF keying to modulate the power supply to the implants for one-way data communication (Fig. 3a). Specifically, sequences of pulses are sent to the device to select a stimulation state saved in the nonvolatile memory on the device (Fig. 3b). Combinations of 90 and 130 ms pulses which represent a logic 0 and 1, respectively are utilized to select a device and its program, that includes specific frequency, duty cycle, and probe selection of the optogenetic stimulus. Figure 3b outlines the basic wireless control principle in which the resonant magnetic field is modulated, indicated by an envelope of a pickup coil (black colored trace). The resulting supply voltage at the μC on the implant follows the power supply (orange trace graph), which is evaluated by internal timers and stores the results in nonvolatile memory which, after completion of the byte code, can be evaluated to set the stimulation state. Resulting μ-ILED timing (blue traces) can be used for a variety of stimulation states. The protocol in the top half set of graphs of Fig. 3b sets the device to operate at a frequency of 30 Hz, a duty cycle of 5%, and with left injectable probe on. The same device in the bottom half set of graphs is set to 10 Hz, a duty cycle of 15%, with the right probe on. All devices were set to operate at 10 mW/mm$^2$, a graph of operating intensities and required electrical power[1,4] can be found in Supplementary Information Fig. S9.

A detailed look at the protocol and subsequent state selection is shown in Fig. 3c. Here, 2 bits are allocated to device selection (four directly addressable devices) followed by 4 bits for frequency selection (16 states), 4 bits for duty cycle selection

(16 states), and 2 bits for probe selection (left, right, and dual probe). The data were modulated with ON/OFF keying using a commercial RF power source (Neurolux Inc.) and custom timing hardware. Multiple devices in the same magnetic field may communicate with the system as shown in Fig. 3d and Supplementary Movie SV1. Mechanical layout of the stimulators was engineered to allow for maximum flexibility during implantation to enable targeting a large amount of brain regions without implant redesign. This is accomplished by stretchable serpentines linking the injectable probe and the device body. Figure 3e demonstrates this in a photographic image where the injectable probe is stretched substantially. The design is validated with finite element simulation (details in Methods section and Supplementary Information Figs. S10, 11). The inset in Fig. 3e shows a displacement of 1.2 mm (24% strain), resulting in a maximum strain in the copper layer of 4% which is well below the yield strain of copper, making it sufficiently and repeatably stretchable for implantation in a variety of positions. An example configuration of the dual probes is shown in Fig. 3f, where the probes are articulated to specific areas of the brain.

**Advanced power management and data communication strategies**. Sensing and stimulation capabilities can expand the utility of implants dramatically, however wireless communication can be a major hurdle due to relatively high-power consumption and footprint requirements. Overcoming these challenges for highly miniaturized devices for the use in songbirds is critical to enable advanced interrogation and modulation capability. To demonstrate the sensing capabilities of the presented platform, we have included thermography capability with mK resolution. The sensing modality enables the observation of physiological baselines such as the circadian rhythm, song tempo, and mating behavior[20]. The low noise and high fidelity of this analog front end also showcases the capability to integrate sensing modalities that require low noise analog electronics that are stable over the duration of the device and animal model lifetime.

A detailed layout of the thermography modality is shown in Fig. 4a. The layout was carefully considered to ensure high sensor fidelity and exclusion of impact of surrounding components on the sensing results. The NTC thermistor, shown in green, was placed in a location to prevent influence of parasitic heating caused by the μC, HF rectifying diodes, Zener protection diodes, and traces that route from these components. Measurements of steady state temperatures of these components in air can be found in Supplementary Information Fig. S12. Corresponding finite element analysis (FEA) validate measurements in air and simulate temperatures when implanted. The results indicate that device components do not influence the recorded temperature or affect the surrounding tissues. Increases in temperature during operation are <0.55 °C at hotspots on the device and <0.005 °C at the NTC sensor. An electrical schematic of the analog front end is shown in Fig. 4b. Here, the NTC thermistor (100 kΩ) is placed in a Wheatstone bridge configuration and the resulting voltage is amplified with a low power operational amplifier (117x gain). A calibration curve for corresponding ADC values and temperatures is provided in Fig. 4c over a dynamic range of 7.37 K indicating a resolution of 1.8 mK and accuracy of ±0.097 K when comparing against a digital thermocouple thermometer (Proster). The thermographic recording capabilities of the device can be used for a range of applications including chronic measurements of sleep–wake cycles in animal subjects as well as recording millisecond-level temperature changes as shown in Supplementary Information Fig. S13.

Data uplink is achieved via IR digital data communication. This mode of data uplink has been successfully implemented in

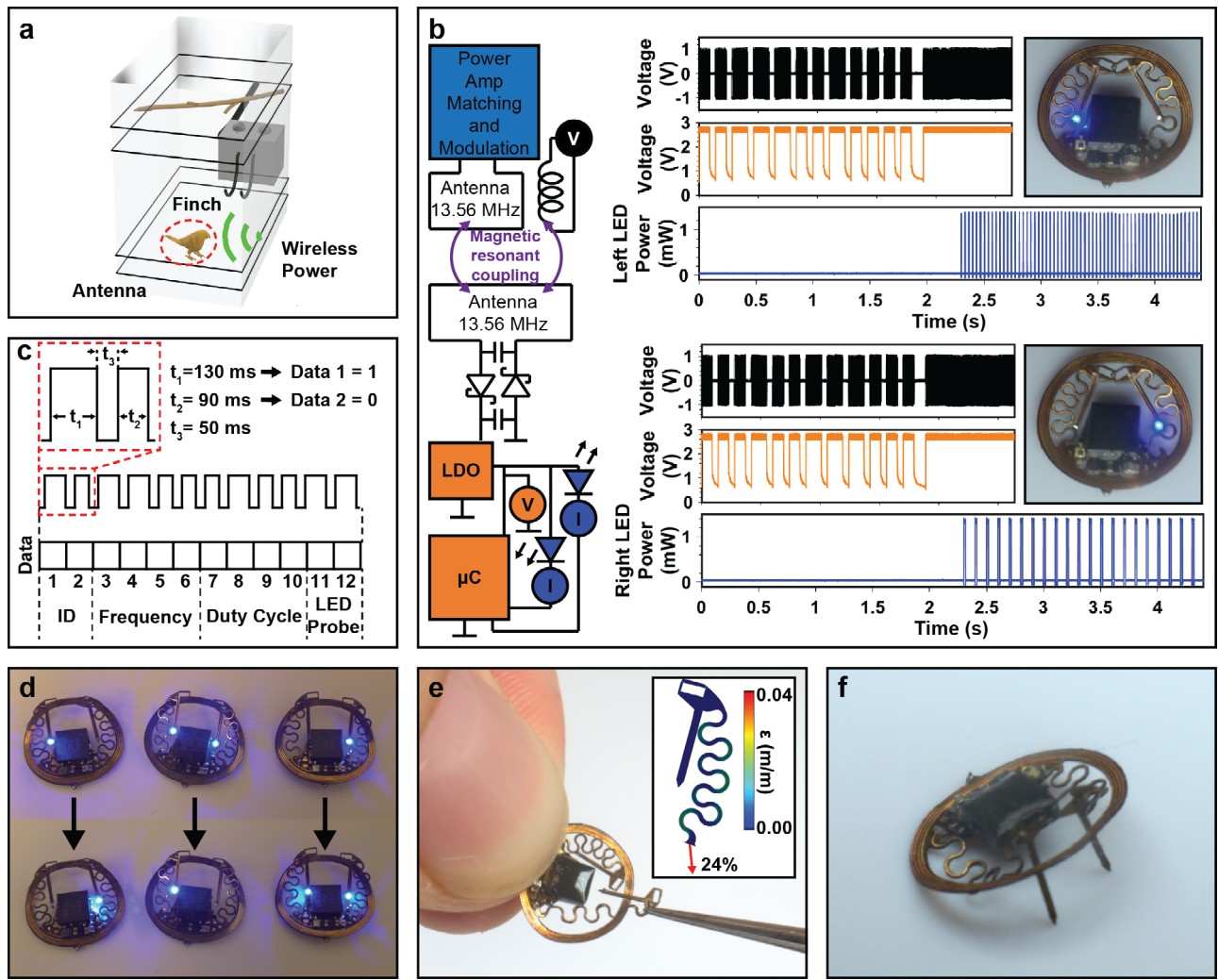

**Fig. 3 Digital modulation schemes for highly miniaturized optogenetic stimulation tools for birds. a** Rendering of experimental arena showing power delivery to animal. **b** Modulation principle with simplified electronic diagram. Graphical representation of pickup coil voltage (black trace), µC voltage (orange trace), and power across µ-ILED (blue trace) to program device for stimulation of left probe at 30 Hz and 5% duty cycle (top set of graphs) and stimulation of right probe at 10 Hz and 15% duty cycle (bottom set of graphs). **c** Protocol for state selection using 12 bits of data to select device, frequency, duty cycle, and stimulation probe. **d** Photographic image of three devices switching states wirelessly. **e** Photographic image of device serpentine stretched in the elastic regime with inset of finite element simulation indicating displacement of 1.2 mm (applied strain of 24%) and observed strain in the copper layer of 4%. **f** Example configuration of injectable dual probes for targeting specific areas of the finch brain.

rodents[2] and is chosen here because of its ultra small footprint (0.5 mm$^2$) and low number of peripheral component needed. Unlike previous studies with rodents, energy throughout the arena is limited and poses a challenge for continuous data uplink.

The simplified electrical schematic of the multimodal optogenetic stimulator and thermography device is shown in Fig. 4d. To enable optogenetic stimulation and data communication on a device with highly miniaturized footprint and therefore limited energy harvesting performance, advanced energy management is required. This is achieved via a capacitive energy storage that holds sufficient energy to support events with power requirements that surpass the harvesting capability. Specifically, this is achieved by a capacitor bank (6 x 22 µF capacitors with a maximum energy storage of 481 µJ and a footprint of 1.5 mm$^2$), the capacitive storage is charged to 5.6 V during events where the power consumption is low, such as µC sleep phases in between stimulation, digitalization, or data uplink, and energy can be withdrawn during high powered events such as data uplink. The charge state of the capacitor bank is indicated in the black voltage trace in Fig. 4d where a data uplink event is recorded. Initially, µC

power consumption (orange trace) of 30 mW during sampling of the ADC and writing of the EEPROM is recorded (green background), followed by the data uplink event which requires a peak power requirement of 19.12 mW and an average power of 8.83 mW (violet background). To achieve lower capacitor bank size and therefore a highly miniaturized footprint, advanced data uplink management is introduced to keep the sending event short. Data uplink events are shortened by splitting data packages into two 8-bit fragments which are recombined to 12 bits at the receiver. This results in a sending event length of 16.9 ms, which requires an energy of 149 µJ, which is less energy than the capacitive energy storage holds (details of overall device power requirements in Supplementary Information Fig. S14). This buffer capability is visible during the voltage drop at the capacitive energy storage during sending events retaining the regulated system voltage and allowing for stable operation even in environments where harvested power does not meet demand on the device. The resulting capability to momentarily store harvested energy enables operation in experimental arenas that extend to volumes that provide lower average power than peak

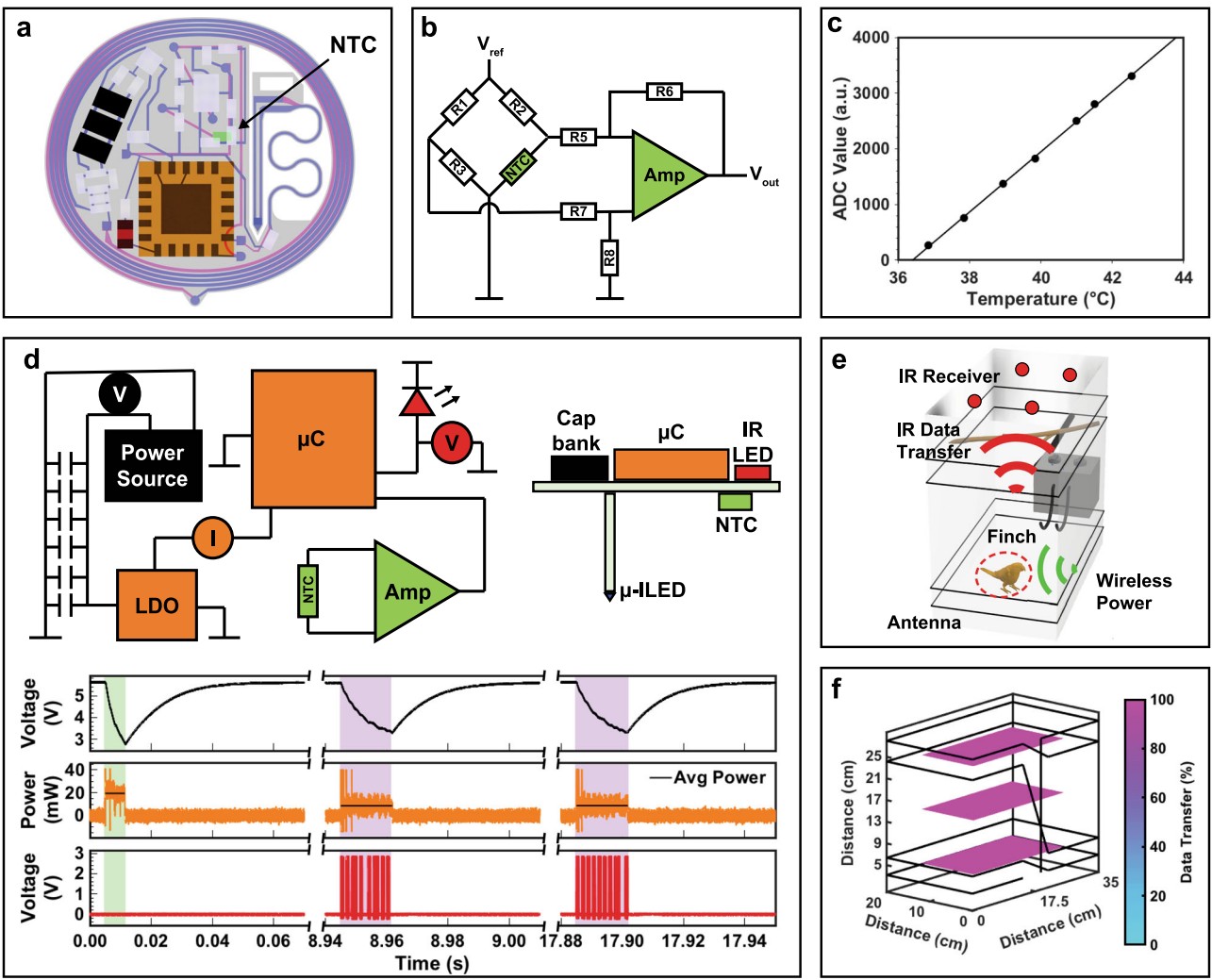

**Fig. 4 Multimodal optogenetic stimulator and thermography device characterization. a** Layout of multimodal optogenetic stimulator and thermography device. **b** Electrical diagram of analog front end, indicating location of resistors, negative temperature coefficient (NTC) sensor, amplifier (Amp), reference voltage ($V_{ref}$), and output voltage ($V_{out}$). **c** Calibration curve of ADC values and temperatures with 1.8 mK resolution. **d** Simplified electronic diagram (top left) and side view schematic of multimodal optogenetic stimulator and thermography device (top right) with voltage measurement of capacitor bank (black trace), power measurement of μC (orange trace), and voltage measurement of IR LED (red trace) (bottom). Green highlighted area indicates sampling of the ADC and writing of EEPROM and violet highlighted area indicates sending of 8-bit fragment. **e** Rendering of experimental arena showing data uplink and power delivery to animal with implanted device. **f** Spatial distribution of uplink stability.

power demand on the device, significantly increasing experimental arena volumes.

A typical setup is illustrated in Fig. 4e, where four receivers are placed at a height of 35 cm and receive signals generated by the implant to enable beyond line of sight data uplink from the test subject[21]. The resulting uplink performance in this experimental paradigm are shown in Fig. 4f and indicate stable data rates with no dropouts. Multimodality of the devices is demonstrated in Supplementary Movie SV2 showing two devices recording temperature and switching optogenetic stimulation protocols simultaneously. For the first time, multiple wireless and battery free devices capable of simultaneous recording and stimulation are demonstrated, enabling control over multiple subjects in the same experimental enclosure.

The result of the miniaturization efforts is a device footprint and outline that allows for facile subdermal implantation. Figure 5a shows a micro-CT scan of the multimodal optogenetic stimulator and thermography device in axial (left) and sagittal/coronal (right) orientation. The device is not visible from the outside and the low weight and profile result in minimal impact

on the subject. This is quantified in subject behavioral experiments. Figure 5b shows heat maps displaying subject behavior over a 14 h time period mapped on a schematic of the experimental enclosure. The spatial position pattern of the bird before and after device implantation is not qualitatively affected. Quantitative indication of minimal impact is evident when computing the distance traveled during the experiment, which shows similar activity before and after device implantation. Impact of the magnetic field on the subjects is minimal as shown by similar studies that use magnetic fields as a power source[1]. Sound emitted by the setup was investigated by recording sound levels and analyzing frequency components in an empty experimental chamber. There was no measurable noise or change in noise with the system active (Supplementary Information Fig. S15).

Proof of concept stimulation capabilities of the device are tested by targeting Area X, a song-dedicated basal ganglia brain nucleus in adult male zebra finches. We use viral delivery pathways established in a prior study by Xiao et al. using adeno-associated virus expressing human channel rhodopsin into the

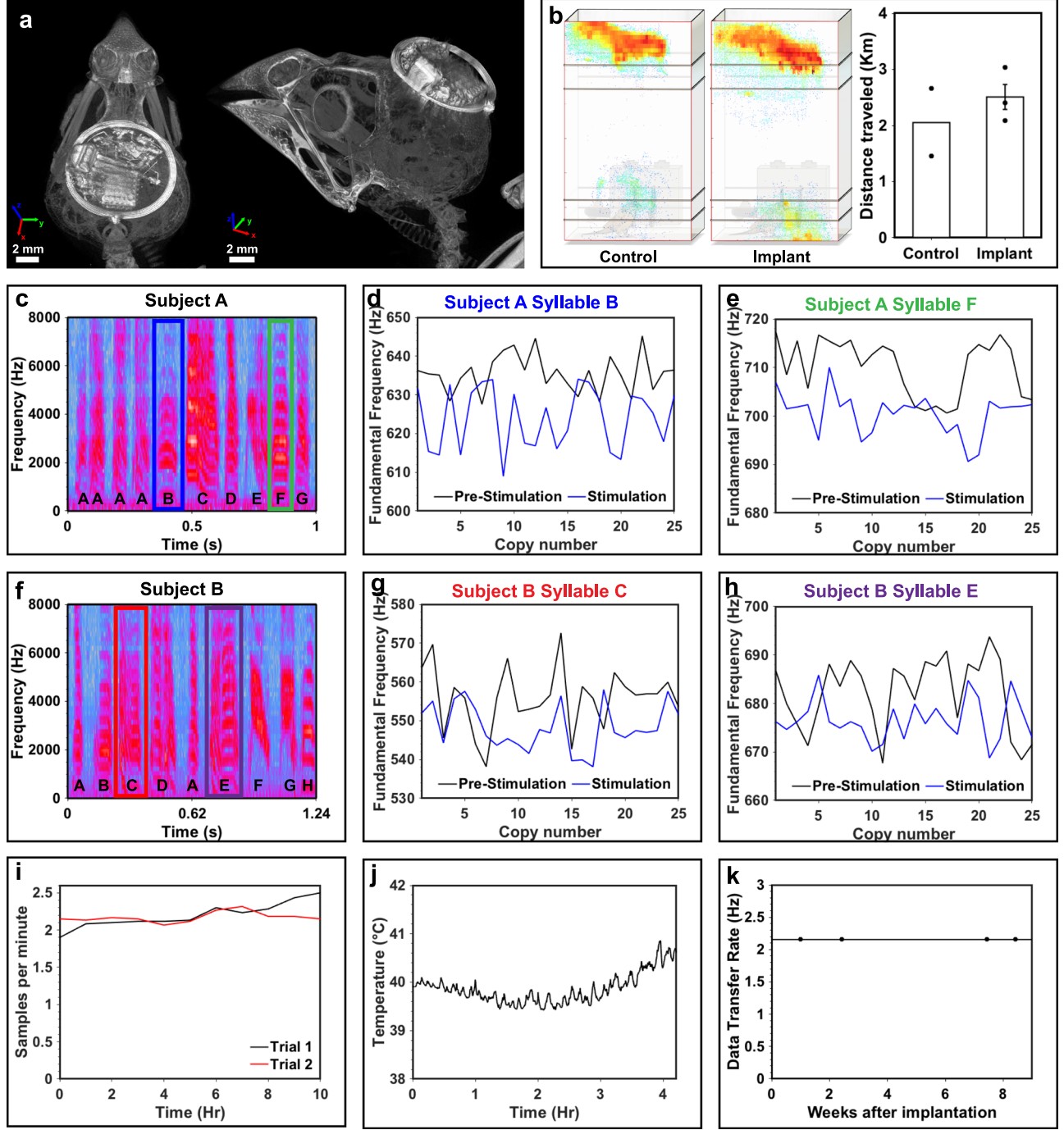

**Fig. 5 Characterization of device platform in freely behaving birds. a** Axial CT slice of the multimodal optogenetic stimulator and thermography (left) and 3D rendered CT (right). **b** Heat maps showing animal model position over a 14-hour period imposed on a 3D rendering of the experimental arena for control and implanted subject (left). Measurement of distance traveled presented as mean values ± SEM for control ($n = 2$) and implanted ($n = 3$) subjects (right). **c** Spectrogram of bird song motif from finch subject A. The motif sequence of individual syllables is denoted by letters A–G. **d** Fundamental frequency ($f_o$) scores of subject A and syllable B pre-stimulation (black line) and post stimulation (blue line) over a 30-minute period and 25 copies. **e** Fundamental frequency ($f_o$) scores of subject A and syllable F pre-stimulation (black line) and post stimulation (blue line) over 25 copies. **f** Spectrogram of bird song motif from finch subject B. The motif sequence of individual syllables is denoted by letters A–H. **g** Fundamental frequency ($f_o$) scores of subject B and syllable C pre-stimulation (black line) and post stimulation (blue line) over 25 copies. **h** Fundamental frequency ($f_o$) scores of subject B and syllable E pre-stimulation (black line) and post stimulation (blue line) over a 15 min period and 25 copies. **i** Uplink rate of the multimodal optogenetic stimulator and thermography device for two 10 h trials. **j** Temperature measurement over 4 h with the multimodal optogenetic stimulator and thermography device. **k** Long term measurements of device uplink rate indicating stable data communication over the course of 9 weeks.

ventral tegmental area (VTA) and taken up by dopaminergic neurons whose axons project to Area X. Prior studies show that continuous optogenetic stimulation of this VTA to Area X pathway over multiple days in tethered adult male zebra finches can alter their ability to pitch shift individual syllables within their songs in a learning task[11]. Here, we use their same viral vector and targeting strategy (Supplementary Information Fig. S16) to stimulate Area X unilaterally over a single, short session (15–30 min) and elicit pitch shifts to demonstrate proof of principle. Stimulation parameters of 20 Hz and 15% duty cycle were used during the experimental sessions. Histological assessment confirms opsin expression and targeting of the probe (Supplementary Information Fig. S16). Song behavior was recorded and analyzed just prior to and throughout the stimulation period (spanning 15–30 min). The basic unit of the bird's song is a motif and is displayed as a spectrogram where individual syllables within the motif are identified based on their structural characteristics (Fig. 5c, f). Here, measurements of fundamental frequency ($f_o$, pitch) are only made from syllables within the song that have a clear, uniform harmonic structure, as per established criteria in the field[22] (Supplementary Information Figs. S17, 18, and 19). Each syllable is then scored across 25 consecutive song renditions to examine millisecond by millisecond changes in $f_o$ (see Methods). Song analysis reveals that across multiple subjects, a statistically significant shift (downward-Subjects A, B or upward-Subject C) in the $f_o$ is detected with stimulation (Fig. 5d, e, g, h). Specifically, stimulation-induced $f_o$ changes compared to pre-stimulation values are shown for Subject A (syllables B, C, D, F, G), Subject B (C, E, but not B), and Subject C (D, E) (Supplementary Information Table ST1/Supplementary Information Fig. S20).

Device operation for multimodal devices with thermography capabilities features stable operation across an extended time period following advanced energy management described above. The resulting system stability in freely moving subjects is shown in Fig. 5i. Sampling rate stays stable over two trials with 10 h of recording time. An example of temperature recording capability is shown in Fig. 5j where a section of 4 h recording shows the temperature minimum in the circadian rhythm of a songbird, indicated by the initial decrease followed by upward trend in temperature consistent with circadian fluctuations when the bird is resting[20].

Device function is checked via link stability measurements that consist of 5-minute-long measurements in a 20-cm-diameter circular experimental arena performed periodically with data rates indicating stable device performance over the course of 9 weeks after implantation when experiments are terminated (Fig. 5k).

## Discussion

Device performance indicate a robust platform that enables the long term and high-fidelity readout of analog sensors with high precision and accuracy coupled with multimodal optogenetic stimulation with a footprint (50.62 mm$^2$ dual optogenetic probe device and 61.05 mm$^2$ multimodal optogenetic stimulator and thermography device), volume (15.19 mm$^3$ dual optogenetic probe device and 19.25 mm$^3$ multimodal optogenetic stimulator and thermography device) and weight (44 mg dual optogenetic probe device and 84 mg multimodal optogenetic stimulator and thermography device) that is sufficiently small for subdermal implantation in passerine birds. Successful optical modulation of songs in zebra finches with minimal impact on subject behavior offers advanced study of flying subjects. The implant function is enabled by advanced concepts for highly miniaturized battery free electronics. Specifically, primary antenna designs guided by deep

neural network analysis of subject video to maximize transmission, advanced protocols for multimodal optogenetic stimulation control, and advanced energy management that strategically uses energy saved in local miniaturized capacitive energy storage to enable data uplink function in experimental enclosures suitable for birds. The combination of these advances provides a toolbox towards the device design for investigation platforms in flying subjects that substantially expands the operational conditions of wireless, battery free and subdermally implantable modulation, and recording tools for the central nervous system.

The current device embodiment features thermal recording capabilities highlighting the ability to include a variety of sensors. Examples for possible future recording capabilities include photometric probes for cell specific recording which have been demonstrated in rodent subjects[2] and electrophysiological recording to capture non cell specific firing activity. The advances presented in this work provide foundational design approaches to enable tools for studies in songbirds for complex social interactions and chronic changes in song and behavior.

## Methods

**Device fabrication**. Flex circuits were composed of Pyralux AP8535R substrate. Top and bottom copper layers (17.5 μm) on a substrate polyimide layer (75 μm) were defined via direct laser ablation (LPKF U4). Ultrasonic cleaning (Vevor; Commercial Ultrasonic Cleaner 2L) was subsequently carried out for 10 min with flux (Superior Flux and Manufacturing Company; Superior #71) and 2 min with isopropyl alcohol (MG Chemicals) and rinsed in deionized water to remove excess particles. Via connections were manually established with copper wire (100 μm) and low temperature solder (Chip Quik; TS391LT). Device components were fixed in place with UV-curable glue (Damn Good 20910DGFL) and cured with a UV lamp (24 W) for 10 min. The devices were encapsulated with a parylene coating via chemical vapor deposition (CVD) (14 μm).

**Electronic components**. Components were manually soldered onto device with low temperature solder (Chip Quik; TS391LT). The rectifier was composed of two Schottky diodes, tuning capacitors (116.2 pF), and a 2.2 μF buffer capacitor. A Zener diode (5.6 V) provided overvoltage protection. A low noise 500 mA low-dropout regulator with fixed internal output (Fairchild FAN25800; 2.7 V) stabilized power to the μC. Six 22 μF capacitors in parallel served as capacitive energy storage to buffer high energy events. Small outline μC (AT-Tiny 84A 3 mm x 3 mm; Atmel) were used to regulate timing μ-ILED activation, readout and digitalization of the NTC, and IR communication. Tiny AVR programmer and USB interface were used to upload software onto the μC. Blue μ-ILED (CREE TR2227) current was set with a current limiting resistor (270 Ω) to control irradiance of the blue LED stimulation. An operational amplifier (Analog Devices ADA4505-1) was used in a differential configuration to readout the resistance of the NTC which was then digitized by the μC. A 0402 IR LED was used to transmit the modulated digital signal (57 kHz).

**RF characterization**. Secondary antenna performance was empirically tested by varying tuning capacitors to produce the lowest voltage standing wave ratio at 13.56 MHz during reflection testing with a reflection bridge (Siglent SSA 3032X). Characterization of the secondary antenna was carried out with a cage in a Helmholtz-like configuration at the center of x-y coordinates and a height of 25 cm from the cage floor with an input power of 10 W. Characterization of primary antennas were carried out with standard (22 AWG) wire wrapped around a custom-built arena (35 cm x 35 cm x 20 cm). An auto-tuner (Neurolux) system was used to tune the cage to a radiofrequency of 13.56 MHz. The power distribution of the primary antenna was characterized by measuring the voltage produced by the primary antenna and a shunt resistor in series at a variety of equidistantly spaced locations at 5, 9, 13, 17, 21, and 25 cm from the cage floor.

**Electrical characterizations**. Electrical properties of the state selection for dual optogenetic probe device and multimodal optogenetic stimulator and thermography device were determined with a current probe (Current Ranger Low-PowerLab) and oscilloscope (Siglent SDS 1202X-E). Alternating voltage of the cage was determined by measuring the voltage of a secondary antenna without a rectifier tuned to 13.56 MHz placed in the magnetic field (pickup coil). Voltage at the μC of the device was determined by measuring the voltage produced after the rectifier and 2.7 V LDO with an oscilloscope. Power to the μ-ILED was measured by passing the current entering the μ-ILED through a current probe (Current Ranger Low-PowerLab). Power harvesting capabilities of the device can be found in Supplementary Information Figs. S7 and S8. Voltage measurements at the capacitor bank during digitalization and sending events of the thermal sensing device were carried out by powering the device with a custom battery powered power supply in series

with a current limiting resistor (900 Ω) to match the power input from the RF field, and subsequently measured with an oscilloscope. The power consumed before the LDO at input voltages that represent the capacitor bank range is shown in Supplementary Information Fig. S14. Voltage measurements of the IR LED during sending events were measured with oscilloscope (Siglent SDS 1202X-E). Power to the μC was determined by passing the current supplied by the LDO to the μC through a current probe (Current Ranger LowPowerLab).

**CT imaging**. Micro-CT imaging was performed on post mortem skull preserved in formalin. Images of the finch were acquired using a Siemens Inveon μ-CT scanner. "Medium-high" magnification, an effective pixel size of 23.85, 2 x 2 binning, with 720 projections made in a full 360 degree scan, along with an exposure time of 300 ms were used. A peak tube voltage of 80 kV and a current of 300 μA was used to obtain the left image of Fig. 5a and a peak tube voltage of 65 kV and a current of 400 μA was used to obtain the right image of Fig. 5a. Reconstruction was done using a Feldkamp cone-beam algorithm.

**Mechanical simulations**. Ansys® 2019 R2 Static Structural was utilized for static-structural FEA simulations to study the elastic strain in both the single and bilateral serpentines when stretched. The components of both devices, including the copper traces, PI, and parylene encapsulation layers were simulated in accurate layouts. The models were simulated using Program Controlled Mechanical Elements, the resolution of the mesh elements being set to 7 with a minimum edge length of 5.813 μm, and at least two elements in each direction in each body to ensure mesh convergence. The Young's modulus ($E$) and Poisson's ratio ($v$) were $E_{PI} = 4$ GPa, $v_{PI} = 0.34$[23]; $E_{CU} = 121$ GPa, $v_{CU} = 0.34$[24]; $E_{Parylene} = 2.7579$ GPa, $v_{Parylene} = 0.4$[25]. For both probes, a fixed support was added to the faces marked B in Supplementary Information Figs. S10 and S11. Strains for the single and bilateral serpentines were applied using a displacement as the load on the faces marked A and in the direction of the arrows shown and are 55% (2.1 mm) and 24% (1.2 mm), respectively.

**Thermal simulations**. COMSOL ® Multiphysics Version 5 was used to create a finite element model to simulate thermal impact of device operation. The models were used to determine steady state temperatures of the device after 500 s of operation. Major heat sources were simulated by using the μC, rectifier, LDO, amplifier, μ-ILED, and IR LED as heat sources. Electrical components, copper traces, and PI were simulated in with component topologies and accurate layout. The mesh was generated with a minimum element size of 0.181 mm and maximum of 1.01 mm. Simulations were set up using natural convection in air with an initial temperature of 22 °C to reflect benchtop experiments of device heating, operation in vivo was simulated with PBS as surrounding medium with an initial temperature of 39 °C to mimic average body temperature of subject. Thermal input powers of each simulated component were set as follows: μC 1 mW; LDO 2 mW; rectifier 10 mW; IR LED 0.5 mW; μ-ILED 0.5 mW; and resistors 0.05 mW. The following thermal conductivity, heat capacity, and density was used for each simulated material: Capacitors: 3.7 W m$^{-1}$ K$^{-1}$, 0.58 J kg$^{-1}$ K$^{-1}$, and 2500 kg m$^{-3}$; Dies of μC, μ-ILED, IR LED, LDO: 34 W m$^{-1}$ K$^{-1}$, 678 J kg$^{-1}$ K$^{-1}$, and 2320 kg m$^{-3}$; Resistors and outer casing of dies of μC, μ-ILED, IR LED, LDO: 0.25 W m$^{-1}$ K$^{-1}$, 1000 J kg$^{-1}$ K$^{-1}$, and 1350 kg m$^{-3}$; Copper traces: 400 W m$^{-1}$ K$^{-1}$, 385 J kg$^{-1}$ K$^{-1}$, and 8700 kg m$^{-3}$; PI: 0.2 W m$^{-1}$ K$^{-1}$, 1100 J kg$^{-1}$ K$^{-1}$, and 1300 kg m$^{-3}$; Water: 0.6 W m$^{-1}$ K$^{-1}$, 4180 J kg$^{-1}$ K$^{-1}$, and 1000 kg m$^{-3}$; Air: 0.026 W m$^{-1}$ K$^{-1}$, 1000 J kg$^{-1}$ K$^{-1}$, and 1.2 kg m$^{-3}$.

**Video tracking and motion analysis**. Videos were recorded with two cameras (Anivia 1080p HD Webcam W8, 1920*1080, 30 FPS) mounted above and in front of the arena for a bird with implant ($n = 3$) and a bird without implant ($n = 2$). The summarized workflow for the tracking steps is available in Supplementary Information Fig. S3. Tracking of the head's position for both top-view and side-view videos was performed using DeepLabCut Version 2.2.b6[18]. A distinct body feature (beak of the bird) was chosen for the tracking. Training was performed individually for each camera view. The training session of the model was accomplished with 13 min of recorded video. The frame extraction rate of 1 frame/second was used to capture 780 frames from each video. The training was computed with a High-Performance Computer (University of Arizona HPC) with 200,000 iterations for each camera view. Software was set for tracking 14-hour video for both camera views. The results of the tracking session were extracted in Excel format containing the X and Y coordinates and the confidence value (Likelihood) of each data point. The data points with the confidence value greater than 99% were used for heat maps and analysis and were then inserted into SimBA (version 1.2) for the evaluation of the total distance traveled[26] as well as for the trace tracking plots (Supplementary Information Fig. S4). The results of the tracking were used to draw the heat maps of each camera view with MATLAB (Version R2020a)[27].

**Animal experiments**. Implanted birds ($n = 3$) were acclimated in the testing arena prior to experiments. Song recordings and stimulation experiments were conducted in a bird cage (35 cm x 35 cm x 20 cm) with perches and water placed inside and housed within a sound attenuation chamber (Eckel Noise Control Technologies,

Cambridge, MA). Birds were recorded and song analyzed from a 15 min period before stimulation and then over a 15–30 min period of stimulation with 10 W input power into the system. Arenas were cleaned and water was changed before each experiment. For temperature sensing experiments with a separate group of finches ($n = 3$), data uplink was established with four IR receivers placed at 35 cm from the base of the arena and connected to a computer with Arduino software. Data communication was recorded for 10 h with an input power of 10 W into the system.

**Subjects and surgical procedures**. All animal use was approved by the Institutional Animal Care and Use Committee at the University of Arizona. Adult male zebra finches ($n = 8$) between 120 and 300 days post-hatch were used in this study. Male finches were moved to sound attenuation chambers (Eckel Industry, Inc., Acoustic Division, Cambridge, MA) under a 14:10 h light: dark cycle and acclimated to their new housing for at least several days prior to surgery. Cameras (Anivia 1080p HD Webcam W8, 1920*1080, 30 FPS) in the chamber recorded the birds' physical movements using iSpy software. Surgery was conducted on isoflurane-anesthetized birds with the analgesic lidocaine injected subcutaneously under the scalp to minimize discomfort prior to device implant. The optogenetic device was implanted subdermally and affixed to the skull by surgical glue. The device consisted of a wireless energy harvesting module and an extendable, articulated stimulating probe with a blue 465 nm light emitting diode at the tip (μ-ILED, CREE TR2227). The stimulating probe was extended unilaterally into song-dedicated brain nucleus Area X using stereotaxic coordinates starting from the bifurcation of the mid-sagittal sinus (40° head angle, 3.5 mm rostral-causal, 1.62 mm medio-lateral, 3.1 mm depth). In the same surgery, finches received a bilateral injection of 700 nl of an adeno-associated virus (AAV) driving human channel rhodopsin (AAV1-CAG-ChR2-ts-HA-NRXN1-p2a-EYFP-WPRE from T. Roberts, U. Texas Southwestern Medical Center) targeting the ventral tegmental/substantia nigra complex (VTA/SN$_C$) that sends nerve projections into Area X (37–40° head angle, 1.65 mm rostral-causal, 0.27–0.40 mm medio-lateral, 5.8–6.33 mm depth). Virus was loaded into a glass electrode pulled pipette fitted into a Nanoject II pressure-injector and back-filled with mineral oil then injected at a rate of 27.6 nl/injection every 15 s for a total of 700 nl per midbrain hemisphere. Prior work established the efficacy of this AAV in the optical excitation of the VTA to Area X pathway by blue light (465 nm)[11] and the subsequent shift in syllable pitch. After 5 min, the pipette was slowly retracted and the tip visually inspected for clogging. Birds were then returned to sound chambers following post-operative monitoring and allowed to recover for a week. Sulfatrim antibiotic was provided in the drinking water. Stimulation was performed between 3 to 4 weeks post-surgery. In two finch subjects, the device implant was affixed to the skull only without the injectable stimulating probe, in order to collect information about scalp temperature (Fig. 5i–k).

**Song recording and analysis**. Morning song from lights-on was acquired from individually housed male finches using Shure 93 lavalier condenser omnidirectional microphones. Songs were digitized (PreSonus Audiobox 1818 VSL, 44.1 KHz sampling rate/24 bit depth, Niles, IL) and recorded by the freeware program, Song Analysis Pro (SAP, http://soundanalysispro.com/)[28]. Spectrograms were then viewed in SAP for further analysis and figures displayed using Audacity (https://www.audacityteam.org/). Zebra finch song consists of a sequence of repeated syllables that comprise a motif (Fig. 5c, f and Supplementary Information Figs. S17, 18, 19). Using SAP, and a semi-automated clustering program (VOICE)[28,29], wav files for 25 consecutive individual syllables with a harmonic stack component were identified (Supplementary Information Fig. S17: Subject A-Syllables B, C, D, F, G; Supplementary Information Fig. S18: Subject B-Syllables B, C, E; Supplementary Information Fig. S19: Subject C-Syllables D, E). Prior power analyses determined that 25 consecutive syllable copies within a bird are sufficient to detect meaningful differences based on experimental condition[30]. The wav files for 25 harmonic syllables were then run in Matlab version R2014a using the SAP SAT Tools and PRAAT to obtain measurements of fundamental frequency, $f_o$ (pitch) as per prior work[22,31]. Individual $f_o$ values were plotted from 25 consecutive copies just prior to optical stimulation, "Pre-Stimulation" and compared to 25 copies of the same syllable during the "Stimulation" period at 20Hz/15P (Supplementary Information Fig. S20).

**Statistical analysis**. Because the syllable data did not fit a normal distribution, we used the non-parametric Wilcoxon signed-rank test for paired data in which the scores for the same syllable were compared between pre-stimulation and stimulation periods. Significance was set at $p < 0.05$ (IBM SPSS Statistics for Windows version 26, Armonk, NY), and $p$ values are reported in Supplementary Information Table ST1.

**Tissue histology**. Finches that received the AAV injection into VTA and optogenetic device implant into Area X were humanely euthanized with an overdose of isoflurane inhalant and then transcardially perfused with warmed saline followed by chilled 4% paraformaldehyde in Dulbecco's Phosphate Buffer Saline. Fixed brains were cryoprotected in 20% sucrose overnight and then coronally sectioned at 30 μm on a Microm cryostat. Tissue was processed for fluorescent immuno-histochemistry, using a procedure similar to Miller et al. 2015[32]: Hydrophobic borders were drawn on the slides using a pap pen (ImmEdge, Vector Labs)

followed by 3 X 5 min washes in 1X TBS with 0.3% Triton X (Tx). To block non-specific antibody binding, the tissue was then incubated for 1 h at room temperature with 5% goat serum (Sigma-Aldrich #G-9023) in TBS/0.3% Tx followed by 3 x 5 min washes in 1% goat serum in TBS/0.3% Tx. Primary antibodies were incubated in a solution of 1% goat serum in TBS/0.3% Tx overnight at 4 °C. For the VTA/SNc region, a primary rabbit polyclonal antibody was applied (1:500 of Tyrosine Hydroxylase-TH, Millipore Sigma #AB152) to mark dopaminergic cell bodies with a primary mouse monoclonal antibody to detect virus expression via Green Fluorescence Protein (1:100, ThermoFisher 3E6, #A11120). A "no primary antibody" control was performed during initial testing. The following day, sections were washed 5 x 5 min in 1x TBS/0.3% Tx and incubated for 3 h at room temperature in fluorescently conjugated secondary antibodies in 1% goat serum with 1x TBS/0.3% Tx (ThermoFisher 1:1000, goat anti-rabbit 647 #A-21245 for TH and goat anti-mouse 568 #A11031 for GFP). After secondary incubation, sections were washed 3 x 10 min in TBS followed by 2 x 5 min washes in filtered TBS. Slides were then cover-slipped in Pro-Long Anti-Fade Gold mounting medium (Invitrogen, #P36930) and imaged on a Leica DMI6000B with a DFC450 color CCD camera (Leica Microsystems, Buffalo Grove, IL) using the Leica LAS-X version 3.3 software. To assess whether the optogenetic probe induced damage in Area X, tissue sections were processed through a Nissl staining procedure (1% thionin, Fishersci #50520580), cleared in xylene (Fishersci X5-4), mounted in DPX (Sigma-Aldrich #6522), and visualized using a Nikon Eclipse E800 bright field stereoscope connected to an Olympus color CCD DP73 camera and CellSens software.

**Reporting summary**. Further information on research design is available in the Nature Research Reporting Summary linked to this article.

## Data availability
The data used in the plots in this paper and other findings in this study are available from the corresponding authors upon reasonable request. Raw songbird data is available at the University of Arizona repository https://arizona.figshare.com/.

## Code availability
The code used to operate the devices in this paper is available from the corresponding authors upon reasonable request.

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

## Acknowledgements
P.G. acknowledges biomedical engineering department startup funds. J.A. acknowledges the support of the national heart, lung, and blood institute NIH 5T32HL007955-19. The authors thank Neurolux Inc. for loaning us a Neurolux Optogenetics System for this study. The authors thank the Simple Behavioral Analysis team (SimBA) for help in using their software. The authors thank Brenda Baggett for micro-CT usage and acknowledge that research reported in this publication was supported by the National Cancer Institute of the National Institutes of Health under award number P30 CA023074. The authors thank T.F. Roberts of the University of Texas Southwestern Medical Center in Dallas for the viral vector, Kent Clemence for cage construction, and University of Arizona Animal Care. We also thank Mr. Douglas W. Cromey for training and use of the Leica DMI6000 through the University of Arizona, Imaging Cores—Life Sciences North under a Core Facilities Pilot Program awarded to J.E.M. J.E.M. also acknowledges start up funds from the Depts. of Neuroscience and Speech, Language and Hearing Sciences.

## Author contributions
J.A., S.J.M., J.E.M., and P.G. designed research. J.A. performed electrical characterization. S.J.M. performed surgical procedures. J.A. and S.J.M. performed behavioral experiments. S.J.M. and J.E.M. performed immunohistochemistry. J.A., A.A., and A.B. fabricated devices. A.A. performed deep neural network analysis. A.B. performed thermal characterizations. R.P. performed mechanical characterizations. J.A., S.J.M., J.E.M., and P.G. analyzed data. J.A., S.J.M., J.E.M., and P.G. wrote the paper.

## Competing interests
The authors declare no competing interests.
