## [Peer Review File · Nature Communications]

Reviewers' Comments:

Reviewer #1:

Remarks to the Author:

The authors report on a miniaturized implantable wireless battery-free platform capable of controlled optogenetic stimulation and temperature recording in songbirds. In comparison to previous work of the senior author, the main challenge addressed here is how to enable continuous operation of the technology in an animal that moves in a 3d environment rather than in the typical 2d one for rodents. The challenge is overcome by primary antenna design and energy management in the implant, enabling successful operation of the device in the 3d environment. The work addresses an important challenge, it is of high quality, and is well reported. I recommend publishing of the work after the following remarks have been addressed:

- Since similar devices have been reported by the senior author previously, it is important for the reader to understand what is different in this work in comparison to previous work regarding challenges, devices, and applications. The manuscript provides some information on this, but this aspect should be further clarified throughout the manuscript.
- The device can measure temperature and that sensor is placed such that it should not be affected by Ohmic heating from the device. However, there is no information about how large the Ohmic heating is from the device during operation. The authors should provide information on this.
- The authors should comment on the viability and challenges for the technology for larger cages and distances.

Reviewer #2:

Remarks to the Author:

In this paper, Ausra et al describe a small and practical device to optostimulate and record temperature. The physical characteristics of the device make it useful for wireless manipulations in small species such as birds or rodents. The authors highlight the advantages of the device for species that "move in 3D". The technical details are in general well explained, and it would be possible to imagine scenarios in which a wireless and battery free device could be useful. However, the experimental evidence presented is insufficient to fully determine if the design, in its current state, provides efficient optogenetic manipulation of neuronal populations and represents advantages over current methods.

In the following lines I express my main concerns regarding the presentation of the case.

Introduction

1. From the title and throughout the introduction, the general message is misleading. It gives the impression that the device can be used to evaluate and stimulate neural signals, when in its current state it can be used to optostimulate and record temperature only. The title must be modified to accurately describe the capabilities of the device.
2. In the introduction, it is argued that in songbirds, widely used to study vocal learning and production, optical fiber-based methods for manipulation of opsins diminish the natural capacity for flying. However, I could not find arguments that indicate that the limited ability to fly would impact on the mechanisms of vocal learning and production. It is also argued that the current methods require special modifications on the experimental cages and limit the possibility for stimulating multiple subjects simultaneously. However, it is not clear if in the proof of principle experiments presented here, the animals used in fig. 5 were stimulated and recorded simultaneously.
3. It should be clearly specified what the authors refer to "moving in 3D". It's not clear to me why

the authors claim that rodents move in 2D. For instance, there are multiple examples in the literature demonstrating 3D encoding in rats (see a very recent example Grieves RM, Jeffery KJ et al, 2020 in NatComms). I would suggest removing 3D navigation references and perhaps simply refer to the fact that this device would be useful to stimulate flying animals.

4. In the same line, authors argue that one of the main advantages of this technology is the possibility to record from animals that navigate in 3D, but the data collected does not necessarily support the case. The device was not designed to record any variable in 3D, nor the authors performed any stimulation depending on specific areas of the experimental volume. In other words, the strength of the technique is not at all related to 3D but the almost complete freedom of movement. I think that restructuring the introduction would make the study more attractive for the wide field of neuroscience, including primate, rodent and avian species.

Results

1. It would be necessary to provide experimental evidence or at least theoretical justification indicating that the electromagnetic fields produced by the antennas won't alter the behavior of sham animals. For example, how comparable are those electromagnetic fields with the ones used for trans cranial magnetic stimulation?

Additionally, perhaps it would be helpful to include an explanation on the potential auditory noise (e.g. clicks or buzzes) that the whole system produces, especially during the periods of optical stimulation. If auditory noise is present, it could affect or distract the animals during song production.

2. On the stimulation control and characterization. It would be necessary to clarify the exact light power delivered with the LEDs in this configuration. It would also be necessary to report if the device is suitable to deliver different light powers. This is a key feature in optogenetic experiments.

3. Regarding the temperature measurements, is nice to have a thermometer in the head of the animal and perhaps evaluate changes in the temperature along the light dark cycle. But perhaps it would be more useful to have a physiological quantification that its actually related to the optical stimulation. Is the thermistor sensitive enough to detect changes in the time scales of the optical stimulations? Was that measured? It would be helpful to have at least a section dedicated to the potential advantages of having these two features, the way it is presented seems unrelated.

4. My main concern regarding the results section and the paper in general, is the absence of a fundamental experiment demonstrating the possibility to modify neuronal activity. How do we know that the device works properly if we don't have any direct evidence attesting the change of neural activity? The main objective of optogenetic stimulation is to accurately manipulate neural activity, hence its fundamental to provide direct evidence that the new device is actually useful for that objective. For example, in figure 5, authors report a change in particular syllables of the song motif. However, we don't know if those changes are actually related to a change in neural activity or other factors, such a change in luminescence in the animal's skull making the animal distracted. Is this device compatible with electrophysiological quantifications?

5. It's not clear how the statistical analysis was performed, for instance, the syllables showing statistic differences were specifically selected for the comparison or on the contrary, all syllables were analyzed, and the ones reported were the only ones showing statistical differences? In any case, this should be explicitly clarified in results and methods sections, since this is the only evidence attesting for the functionality of the device. It would also be necessary to compare syllables before, during and after stimulations and not only before and during stimulations.

6. It's also noticeable the absence of histological confirmation of infection sites and the potential tissue damage produce by the insertion of the optical devices. Since there's no direct evidence on neural control, this data becomes fundamental to understand the behavioral effects.

The discussion is not actually a discussion but a summary of the results section, therefore it must be rewritten to include the limitations of the current design, the possibilities for improving it or combining it with other techniques for neural interrogations.

Minor comments

In figure 5, it would be useful to include standard deviation indicators in subpanels d-e and g-h. To have a visual comparison of the phenomena, it would also be nice to have spectrograms extracted from stimulated periods.

Would it be possible to know the places where the animals were stimulated? In figure 5b, it's remarkable that the vast majority of the time, the animals remained in the upper part of the cage, but the distance traveled is in the range of kilometers, does that mean that the animals moved a lot on that small upper space? Presented like that, it's not very clear how this device is ideal for the so-called 3D moving species.

Reviewer #3:

Remarks to the Author:

This paper reports the development of a wireless, battery free and multimodal platform that enables optogenetic stimulation and physiological recording in a miniaturized form factor for use in songbirds. Additionally, the design approaches used by authors were able to expand the use of wireless subdermally implantable neuromodulation and sensing tools to species previously excluded from in vivo real time experiments.

The paper is interesting, quite well structured and written, with good results and well discussed. Therefore, I recommend this paper for publication in the Nature Communications edition. Nevertheless, the paper would benefit from a few minor corrections to achieve better clarity to the reader:

- Page 18: "Videos were recorded with 2 cameras (Anivia 1080p HD Webcam W8, 1920*1080, 30 FPS) mounted above and in front of the arena for a bird with implant (n=1) and a bird without implant (n=1)." Why only n=1 was used? Is this statistically significant?
- The same can be observed in page 19: "Implanted birds (n=2) were acclimated in the testing arena prior to experiments". Why was used only n=2?

Response to comments of Referee #1

Reviewer #1 (remarks to the author):

General Comments:

The authors report on a miniaturized implantable wireless battery-free platform capable of controlled optogenetic stimulation and temperature recording in songbirds. In comparison to previous work of the senior author, the main challenge addressed here is how to enable continuous operation of the technology in an animal that moves in a 3d environment rather than in the typical 2d one for rodents. The challenge is overcome by primary antenna design and energy management in the implant, enabling successful operation of the device in the 3d environment. The work addresses an important challenge, it is of high quality, and is well reported. I recommend publishing of the work after the following remarks have been addressed:

Our Response:

We thank the reviewer for these positive comments, positive assessment of impact and insightful comments that we have addressed in detail below.

Comment 1:

Since similar devices have been reported by the senior author previously, it is important for the reader to understand what is different in this work in comparison to previous work regarding challenges, devices, and applications. The manuscript provides some information on this, but this aspect should be further calcified throughout the manuscript.

Our Response:

We thank the reviewer for this comment and agree that novelty and impact of this work should be highlighted more prominently. Novelty introduced in this manuscript can be summarized in three major achievements required for the demonstrations in this work and fundamentally for future technological embodiments.

First, one of the challenges presented here includes the larger arena volume occupied by freely flying animal model as well as smaller device footprint due to the small animal model skull when compared to rodent subjects. As shown in Fig. 2a-e, we address the issue of increased cage volume by introducing a new antenna design technique that utilizes deep neural network analysis of songbird behavior to create primary antennas that guide wireless power towards areas that are predominantly occupied. This capability is complemented by the ability to achieve efficient energy harvesting compared to device size highlighted in Figs. 2f-g and the ability to store the energy with capacitive energy storages that enables seamless operation throughout the volume (Fig. 4d). All of these techniques have not been shown in prior literature and are unique to this manuscript. The techniques are broadly applicable to near field powered implants and will be important to enable advanced device capabilities of this device class.

Second, we could achieve a highly miniaturized electronics footprint, while using all off the shelf components. The layout enables a reduction of occupied footprint by 37.5% in comparison to our prior work. This miniaturization enables the inclusion of advanced multimodal operation, specifically functionality in thermographic recording and optogenetic stimulation in one device. This is the first time that these capabilities have been combined in freely behaving animal models.

Third, we have introduced a new simplex digital communication protocol that enables digital addressing of multiple devices using only standard microcontroller peripherals which is highly beneficial for component count, devices size and power consumption.

To highlight these key advances, we have made substantial modifications to the manuscript and have also included a supplemental video that showcases multimodal operation and multi device addressing.

Modification to the manuscript:

Page 4 we added:

“In this work, we use zebra finches which belong to the family of songbirds, which have an average available head area of 0.825 cm² (Supplemental Information Fig. S1) and typically weigh 14.4 g¹⁵. This is a 10.5% reduction in available head space as compared to mouse animal models¹⁶.”

Citations added:

- (16) M. Kawakami, K. Yamamura, Cranial bone morphometric study among mouse strains. *BMC Evol. Biol.* **8**, 73 (2008).

Page 5 we added:

“In comparison to wireless subdermally implantable and battery free devices for the use in rodents, birds often occupy larger volumes of space. By comparison, standard mouse animal model experimental cage volumes are 45% smaller when compared to standard experimental arenas used for songbirds^{1,7,17}.”

Citations added:

- (1) V. K. Samineni, J. Yoon, K. E. Crawford, Y. R. Jeong, K. C. McKenzie, G. Shin, Z. Xie, S. S. Sundaram, Y. Li, M. Y. Yang, J. Kim, D. Wu, Y. Xue, X. Feng, Y. Huang, A. D. Mickle, A. Banks, J. S. Ha, J. P. Golden, J. A. Rogers, R. W. Gereau 4th, Fully implantable, battery-free wireless optoelectronic devices for spinal optogenetics. *Pain.* **158**, 2108–2116 (2017).
- (7) G. Shin, A. M. Gomez, R. Al-Hasani, Y. R. Jeong, J. Kim, Z. Xie, A. Banks, S. M. Lee, S. Y. Han, C. J. Yoo, J.-L. Lee, S. H. Lee, J. Kurniawan, J. Tureb, Z. Guo, J. Yoon, S.-I. Park, S. Y. Bang, Y. Nam, M. C. Walicki, V. K. Samineni, A. D. Mickle, K. Lee, S. Y. Heo, J. G. McCall, T. Pan, L. Wang, X. Feng, T. Kim, J. K. Kim, Y. Li, Y. Huang, R. W. Gereau, J. S. Ha, M. R. Bruchas, J. A. Rogers, Flexible Near-Field Wireless Optoelectronics as Subdermal Implants for Broad Applications in Optogenetics. *Neuron.* **93**, 509-521.e3 (2017).
- (17) P. Gutruf, V. Krishnamurthi, A. Vázquez-Guardado, Z. Xie, A. Banks, C.-J. Su, Y. Xu, C. R. Haney, E. A. Waters, I. Kandela, S. R. Krishnan, T. Ray, J. P. Leshock, Y. Huang, D. Chanda, J. A. Rogers, Fully implantable optoelectronic systems for battery-free, multimodal operation in neuroscience research. *Nat. Electron.* **1**, 652–660 (2018).

Page 7 we added:

“This approach to modify antenna geometry to tailor power delivery to the subject based on spatial position is ubiquitous, enabling new experimental paradigms.”

Page 8 we added:

“This antenna option outperforms the 10 turn variant at 6.0 V and below significantly, surpassing the 10 turn device by 1.35 mW at the typical operation voltage of the rectifier

at high loads (3.5 V). This optimization enables higher harvested power and improves usable footprint for electronics, sensors and neuromodulation probes.”

Page 12 we added:

“Data uplink is achieved via infrared (IR) digital data communication. This mode of data uplink has been successfully implemented in rodents⁶ and is chosen here because of its ultra small footprint (0.5 mm²) and low number of peripheral component needed. Unlike previous studies with rodents, energy throughout the arena is limited and poses a challenge for continuous data uplink.”

Page 13 we added:

“This buffer capability is visible during the voltage drop at the capacitive energy storage during sending events retaining the regulated system voltage and allowing for stable operation even in environments where harvested power does not meet demand on the device. The resulting capability to momentarily store harvested energy enables operation in experimental arena that extend to volumes that provide lower average power than peak power demand on the device, significantly increasing experimental arena volumes.”

Page 13 we added:

“The resulting uplink performance in this experimental paradigm are shown in Fig. 4f and indicate stable data rates with no dropouts. Multimodality of the devices is demonstrated in Supplemental Movie SV2 showing two devices recording temperature and switching optogenetic stimulation protocols simultaneously. For the first time, multiple devices capable of simultaneous physiological recording and stimulation are demonstrated, enabling complete control over multiple subjects in the same experimental enclosure.”

Additional Videos:

SV2. Demonstration of multimodal operation and multi device addressing.

Comment 2:

The device can measure temperature and that sensor is placed such that it should not be affected by Ohmic heating from the device. However, there is no information about how large the Ohmic heating is from the device during operation. The authors should provide information on this.

Our Response:

We thank the reviewer for this comment and we agree that this information should be quantified. In additional experiments, we evaluate the self heating of the device during operation by mapping heating of the device with a small outline (1.0 x 0.5 x 0.5 mm) NTC SMD thermistor that is placed in various locations to map thermal impact on the devices. We also quantify impact of device heating on components with thermal simulations carried out in air at room temperature and with physiologically relevant thermal environment at subject body temperature. These results indicate no significant parasitic heating at the NTC caused by device operation.

Modification to the manuscript:**Page 11 we added:**

“The NTC thermistor, shown in green, was placed in a location to prevent influence of parasitic heating caused by the μC , HF rectifying diodes, and Zener protection diodes. Measurements of steady state temperatures of these components in air can be found in Supplemental Information Fig. S12. Corresponding FEA validate measurements in air and simulate temperatures when implanted. The results indicate that device components do not influence the recorded temperature or affect the surrounding tissues. Increases in temperature during operation are $< 0.55\text{ }^\circ\text{C}$ at hotspots on the device and $< 0.005\text{ }^\circ\text{C}$ at the NTC sensor.”

Page 21 we added:*Thermal Simulations*

COMSOL® Multiphysics Version 5 was used to create a finite-element model to simulate thermal impact of device operation. The models were used to determine steady state temperatures of the device after 500 seconds of operation. Major heat sources were simulated by using the μC , rectifier, LDO, amplifier, $\mu\text{-ILED}$, and IR LED as heat sources. Electrical components, copper traces, and PI were simulated in with component topologies and accurate layout. The mesh was generated with a minimum element size of 0.181 mm and maximum of 1.01 mm. Simulations were set up using natural convection in air with an initial temperature of $22\text{ }^\circ\text{C}$ to reflect benchtop experiments of device heating, operation *in vivo* was simulated with PBS as surrounding medium with an initial temperature of $39\text{ }^\circ\text{C}$ to mimic average body temperature of subject. Thermal input powers of each simulated component were set as follows: μC 1 mW; LDO 2 mW; rectifier 10 mW; IR LED 0.5 mW; $\mu\text{-ILED}$ 0.5 mW; and resistors 0.05 mW. The following thermal conductivity, heat capacity, and density was used for each simulated material: Capacitors:

$3.7 \text{ W m}^{-1} \text{ K}^{-1}$, $0.58 \text{ J kg}^{-1} \text{ K}^{-1}$, and 2500 kg m^{-3} ; Dies of μC , $\mu\text{-ILED}$, IR LED, LDO: $34 \text{ W m}^{-1} \text{ K}^{-1}$, $678 \text{ J kg}^{-1} \text{ K}^{-1}$, and 2320 kg m^{-3} ; Resistors and outer casing of dies of μC , $\mu\text{-ILED}$, IR LED, LDO: $0.25 \text{ W m}^{-1} \text{ K}^{-1}$, $1000 \text{ J kg}^{-1} \text{ K}^{-1}$, and 1350 kg m^{-3} ; Copper traces: $400 \text{ W m}^{-1} \text{ K}^{-1}$, $385 \text{ J kg}^{-1} \text{ K}^{-1}$, and 8700 kg m^{-3} ; PI: $0.2 \text{ W m}^{-1} \text{ K}^{-1}$, $1100 \text{ J kg}^{-1} \text{ K}^{-1}$, and 1300 kg m^{-3} ; Water: $0.6 \text{ W m}^{-1} \text{ K}^{-1}$, $4180 \text{ J kg}^{-1} \text{ K}^{-1}$, and 1000 kg m^{-3} ; Air: $0.026 \text{ W m}^{-1} \text{ K}^{-1}$, $1000 \text{ J kg}^{-1} \text{ K}^{-1}$, and 1.2 kg m^{-3} .

Additional Figures:

Figure S12. Steady state temperature measurements of device components during operation. (a) Schematic of device components indicating location of investigated components. (b) Steady state temperatures for LDO, rectifier, μC , and amplifier during operation (3W RF input into dual loop antenna 8 cm in diameter) in air. (c) Top view steady state thermal FEA analysis of device during operation in air without convection with initial temperature of 22°C . Temperature increases by 9.57°C for rectifier, 7.2°C for LDO, 4.8°C for amplifier, and 6.4°C for μC . (d) Corresponding bottom view steady state

thermal impact analysis. Temperature increases by 4.5 °C for NTC. (e) Top view steady state thermal impact analysis of device during operation in PBS with initial temperature of 39 °C. Temperature increases by 0.510 °C for rectifier, 0.301 °C for LDO, 0.085 °C for amplifier, and 0.085 °C for μ C. (f) Bottom view steady state thermal impact analysis of device during operation in PBS with initial temperature of 39 °C. Temperature increases at the NTC thermal sensor are 0.005 °C.

Comment 3:

The authors should comment on the viability and challenges for the technology for larger cages and distances.

Our Response:

We thank the reviewer for this comment and agree that it is important to consider the extent to which this technology is viable in larger arenas or arenas with unusual dimensions. The technology presented here is capable of changing cage dimensions while maintaining the same volume. Therefore, we show the possibility to operate devices in cages with, for example a long x-axis and characterize the power distribution throughout such an enclosure. We also showcase the possibility to drastically increase the arena volume size by making use of additional RF power sources, allowing for expansion of the experimental arena. This flexibility in coverage of experimental arenas enables a wide variety of setups for freely flying animals.

Modification to the manuscript:**Page 7 we added:**

“This approach to modify antenna geometry to tailor power delivery to the subject based on spatial position is ubiquitous, **enabling new experimental paradigms**. Strategies to preferentially deliver power are demonstrated in Supplemental Information Fig. S5. **It is also possible to change arena dimensions significantly if the same volume is maintained. This is demonstrated in Supplemental Information Fig. S6a where an arena length of 105 cm with cross-sectional dimensions of 15 x 15 cm can sustain device operation. Arenas with increased volumes are also possible by combining multiple RF power sources to achieve coverage for larger volumes (Supplemental Information Fig. S6b-d) with arbitrary shape significantly expanding the utility of the devices to investigate behavior with simultaneous neuromodulation.**”

Additional Figures:

Figure S6. Alternative cage dimensions and sizes. (a) Power distribution at heights of 3, 7.5, and 12 cm from cage floor of a 15 x 15 x 105 cm cage with dual loop antenna and 10 W input power at heights of 4 and 11 cm from the cage floor. (b) Power distribution at heights 3, 6, 15, 27 cm from cage floor of a 20 x 70 x 35 cm cage composed of two antennas at heights of 3, 6, 15, 27 cm from the cage floor with 8 W power to both antennas on. The area in between the cages also receives RF power resulting in the ability to link multiple arenas to build large experimental spaces for freely flying animals. (c) Power distribution at heights 6 and 27 cm from cage floor of a 20 x 70 x 35 cm cage composed of two antennas at heights of 3, 6, 15, 27 cm from the cage floor with left antenna on at 8 W. (d) Power distribution at heights 3 and 15 cm from cage floor of a 20 x 70 x 35 cm cage composed of two antennas at heights of 3, 6, 15, 27 cm from the cage floor with right antenna on at 8 W.

Response to comments of Referee #2

Reviewer #2 (remarks to the author):

General Comments:

In this paper, Ausra et al describe a small and practical device to optostimulate and record temperature. The physical characteristics of the drive make it useful for wireless manipulations in small species such as birds or rodents. The authors highlight the advantages of the device for species that “move in 3D”. The technical details are in general well explained, and it would be possible to imagine scenarios in which a wireless and battery free device could be useful. However, the experimental evidence presented is insufficient to fully determine if the design, in its current state, provides efficient optogenetic manipulation of neuronal populations and represents advantages over current methods.

In the following lines I express my main concerns regarding the presentation of the case.

Our Response:

We thank the reviewer for these comments and agree that the device presented in this work expands the applications of optogenetic tools to small species that can fly. We understand the reviewers concerns on the characterization of the optogenetic modulation and have presented significant new data to address the reviewers concerns.

We would also like to highlight that this work is predominantly focused on characterizing the technology and documenting novel approaches in device engineering and display capabilities of the implants that enable the operation in freely flying subjects. Extensive characterization of the effects of optogenetic neuromodulation in flying songbirds was not previously possible and is the subject of ongoing work in our groups and exhaustive biological and neuroscience studies are out of the scope of this manuscript.

Comment 1:

From the title and throughout the introduction, the general message is misleading. It gives the impression that the device can be used to evaluate and stimulate neural signals, when in its current state it can be used to optostimulate and record temperature only. The title must be modified to accurately describe the capabilities of the device.

Our Response:

We thank the reviewer for the comment and we can see how there could be the impression that the devices are able to record and manipulate neural function. We have revised introduction and abstract for more clarity. Additionally, we have performed experiments that showcase that the device has the ability to record physiology (temperature) and stimulate simultaneously. This is also demonstrated with multiple devices at the same time to highlight the capability to enable multi subject operation.

Modification to the manuscript:**Page 2 we added:**

“Here we report on a wireless, battery free and multimodal platform that enables optogenetic stimulation and physiological **temperature** recording in a highly miniaturized form factor for use in songbirds. The systems are enabled by behavior guided primary antenna design and advanced energy management to ensure stable optogenetic stimulation and **thermography** throughout 3D experimental arenas.”

Page 3 we added:

“Wireless and battery free devices with the ability to optogenetically stimulate and record **temperature** have not been utilized in the context of **flying** species due to challenges in primary antenna design, device miniaturization and digital data communication throughout the typically larger experimental arena volumes required for such subjects.”

Page 13 we added:

“The resulting uplink performance in this experimental paradigm are shown in Fig. 4f and indicate stable data rates with no dropouts. **Multimodality of the devices is demonstrated in Supplemental Movie SV2 showing two devices recording temperature and switching optogenetic stimulation protocols simultaneously. For the first time, multiple devices capable of simultaneous physiological recording and stimulation are demonstrated, enabling complete control over multiple subjects in the same experimental enclosure.**”

Additional Videos:

SV2. Demonstration of multimodal operation and multi device addressing.

Comment 2:

In the introduction, it is argued that in songbirds, widely used to study vocal learning and production, optical fiber-based methods for manipulation of opsins diminish the natural capacity for flying. However, I could not find arguments that indicate that the limited ability to fly would impact on the mechanisms of vocal learning and production. It is also argued that the current methods require special modifications on the experimental cages and limit the possibility for stimulating multiple subjects simultaneously. However, it is not clear if in the proof of principle experiments presented here, the animals used in fig. 5 were stimulated and recorded simultaneously.

Our Response:

We thank the reviewer for this insightful comment. In comparison to the rodent literature, there are not many studies in songbirds that utilize optogenetic approaches. A review of these studies (R1-3), indicates that the researchers did not compare the bird's movement (flying, perching) prior to and during tethering to evaluate impact on motion, song or other physiological factors. Thus, in future work, it will indeed be important to evaluate impacts of tethering for optogenetic devices versus our wireless approach on songbird vocal learning/production. Because of the lack of references in the literature we have modified our statements in the manuscript accordingly. As shown in Fig. 5b and Supplemental Fig. S4, songbirds display a natural behavior of motion throughout their enclosure, both with our devices and in comparison to non-implanted naïve birds.

As shown in Fig. 2h, multiple devices can operate within the same experimental enclosure as close as 1 cm apart. To further illustrate the capability to operate multiple devices while utilizing their multimodal function, we have prepared a video demonstration that showcases simultaneous device operation, including the capability to wirelessly control the optogenetic stimulation parameters of each individual device and capture individual thermography data within one enclosure. A future goal in device design is to record the firing activity of neurons.

We would also like to note that this technology is not limited to studies on vocal learning and production. The ability to deliver optogenetic stimulus to a freely flying subject can be used in a variety of experimental paradigms.

References:

- (R1) Xiao, G. Chattree, F. G. Orosco, M. Cao, M. J. Wanat, T. F. Roberts, A Basal Ganglia Circuit Sufficient to Guide Birdsong Learning. *Neuron*. **98**, 208-221.e5 (2018).
- (R2) W. Zhao, F. Garcia-Orosco, D. Dinh, T. F. Roberts, *Science* **366**, 83-89 (2019) doi:10.1126/science.aaw4226.
- (R3) T. F. Roberts, S. M. H. Gobes, M. Murugan, B. P. Ölveczky, R. Mooney, Motor circuits are required to encode a sensory model for imitative learning. *Nat. Neurosci.* **15**, 1454–1459 (2012).

Modification to the manuscript:

Page 3 we added:

“However, as shown in rodent species, the current methods of stimulation via optical fibers limit animal mobility⁶, induce stress¹², cause mechanical damage¹³, and require advanced cable management¹⁴. These limitations essentially eliminate free flight and multi animal experiments in flying animals.”

Citations added:

- (12) R. P. Kale, A. Z. Kouzani, K. Walder, M. Berk, S. J. Tye, Evolution of optogenetic microdevices. *Neurophotonics*. **2**, 31206 (2015).
- (13) D. Miyamoto, M. Murayama, The fiber-optic imaging and manipulation of neural activity during animal behavior. *Neurosci. Res.* **103**, 1–9 (2016).
- (14) B. B. Land, C. E. Brayton, K. Furman, Z. LaPalombara, R. Dileone, Optogenetic inhibition of neurons by internal light production. *Front. Behav. Neurosci.* **8** (2014).

Page 13 we added:

“The resulting uplink performance in this experimental paradigm are shown in Fig. 4f and indicate stable data rates with no dropouts. Multimodality of the devices is demonstrated in Supplemental Movie SV2 showing two devices recording temperature and switching optogenetic stimulation protocols simultaneously. For the first time, multiple devices capable of simultaneous physiological recording and stimulation are demonstrated, enabling complete control over multiple subjects in the same experimental enclosure.”

Additional Videos:

SV2. Demonstration of multimodal operation and multi device addressing.

Comment 3:

It should be clearly specified what the authors refer to “moving in 3D”. It’s not clear to me why the authors claim that rodents move in 2D. For instance, there are multiple examples in the literature demonstrating 3D encoding in rats (see a very recent example Grieves RM, Jeffery KJ et al, 2020 in NatComms). I would suggest removing 3D navigation references and perhaps simply refer to the fact that this device would be useful to stimulate flying animals.

Our Response:

We thank the reviewer for the comment and agree that the term “moving in 3D” should be clarified. We agree that the claim that rodents move in 2D was oversimplified and have removed this claim. We have removed reference of the device as a tool for subjects “moving in 3D” and have stated that the device in our work is appropriate for optogenetically stimulating flying animals.

Modification to the manuscript:

Page 1 we added:

“While these advantages have been successfully proven in rodents the full potential of a wireless and battery free device can be harnessed with **flying** species, where interrogation with tethered devices is very difficult or impossible.”

Page 2 we added:

“**Flying subjects** have previously been only rarely used for optogenetic modulation⁸ as they are commonly ruled out as species to perform behavioral experiments with due to the lack of optogenetic tools that can support their free movement.”

Citations added:

- (8) E. Hisey, M. G. Kearney, R. Mooney, A common neural circuit mechanism for internally guided and externally reinforced forms of motor learning. *Nat. Neurosci.* **21**, 589–597 (2018).

Page 3 we added:

“Wireless and battery free devices with the ability to optogenetically stimulate and record **temperature** have not been utilized in the context of **flying** species due to challenges in primary antenna design, device miniaturization and digital data communication throughout the typically larger experimental arena volumes required for such subjects.”

Page 5 we added:

“In comparison to wireless subdermally implantable and battery free devices for the use in rodents, **birds often occupy larger volumes of space.**”

Page 16 we added:

“Successful optical modulation of songs in zebra finches with minimal impact on subject behavior offers advanced study of **flying** subjects.”

Page 17 we added:

“The combination of these advances provides a toolbox towards the device design for investigation platforms in **flying** subjects that substantially expands the operational conditions of wireless, battery free and subdermally implantable modulation and recording tools for the central nervous system.”

Comment 4:

In the same line, authors argue that one of the main advantages of this technology is the possibility to record from animals that navigate in 3D, but the data collected does not necessarily support the case. The device was not designed to record any variable in 3D, nor the authors performed any stimulation depending on specific areas of the experimental volume. In other words, the strength of the technique is not at all related to 3D but the almost complete freedom of movement. I think that restructuring the introduction would make the study more attractive for the wide field of neuroscience, including primate, rodent and avian species.

Our Response:

We thank the author for their comment and agree that the device is not meant to record variables related to the 3D location of the animal, but instead is an appropriate tool for modulating neuronal populations and thermography recording of flying species. We have adjusted the text to reflect that we are not recording variables related to the location of the animal by removing phrases that point to the animals moving in 3D.

Modification to the manuscript:

Page 1 we added:

“While these advantages have been successfully proven in rodents the full potential of a wireless and battery free device can be harnessed with **flying** species, where interrogation with tethered devices is very difficult or impossible.”

Page 2 we added:

“**Flying subjects** have previously been only rarely used for optogenetic modulation⁸ as they are commonly ruled out as species to perform behavioral experiments with due to the lack of optogenetic tools that can support their free movement.”

Citations added:

- (8) E. Hisey, M. G. Kearney, R. Mooney, A common neural circuit mechanism for internally guided and externally reinforced forms of motor learning. *Nat. Neurosci.* **21**, 589–597 (2018).

Page 3 we added:

“Wireless and battery free devices with the ability to optogenetically stimulate and record **temperature** have not been utilized in the context of **flying** species due to challenges in primary antenna design, device miniaturization and digital data communication throughout the typically larger experimental arena volumes required for such subjects.”

Page 16 we added:

“Successful optical modulation of songs in zebra finches with minimal impact on subject behavior offers advanced study of flying subjects.”

Page 17 we added:

“The combination of these advances provides a toolbox towards the device design for investigation platforms in flying subjects that substantially expands the operational conditions of wireless, battery free and subdermally implantable modulation and recording tools for the central nervous system.”

Comment 5:

It would be necessary to provide experimental evidence or at least theoretical justification indicating that the electromagnetic fields produced by the antennas won't alter the behavior of sham animals. For example, how comparable are those electromagnetic fields with the ones used for transcranial magnetic stimulation? Additionally, perhaps it would be helpful to include an explanation on the potential auditory noise (e.g. clicks or buzzes) that the whole system produces, especially during the periods of optical stimulation. If auditory noise is present, it could affect or distract the animals during song production.

Our Response:

We thank the reviewer for their comments and agree that the effect of the magnetic field on the subject and other stimuli emitted by the system should be considered. We would like to mention that our system is using frequencies and RF powers that are well within FCC regulation and powers used here are well below thresholds that could induce effects in living subjects. Specifically, we use the ISM band (13.56 MHz) and powers below 10 W (this is commonly used for the readout of RFID tags in libraries for example). The magnetic fields do not introduce heating in tissue because of low absorption at the operation frequency. There are no hot spots caused due to constructive or destructive interference because the system is operating in the near field, and there are no magnetic gradients caused by the field due to its uniformity (as characterized in Fig. 5d, e/Supplementary Fig. S5). The magnetic field created by the devices is not strong enough for magnetic stimulation such as those seen in TMS (our system is continuously operating, TMS coils induce high powered pulses in coils to create magnetic fields several orders of magnitude higher with rapid temporal change in magnetic field to induce depolarization in neuronal populations).

We have investigated the impact of auditory noise produced by the whole system by recording the sound levels and analyze changes in amplitude and the frequency spectrum and can conclude that there is no measurable difference in noise with and without the system active.

Modification to the manuscript:**Page 14 we added:**

“Quantitative indication of minimal impact is evident when computing the distance traveled during the experiment which shows similar activity before and after device implantation. **Impact of the magnetic field on the subjects is minimal as shown by similar studies that use magnetic fields as a power source¹. Sound emitted by the setup was investigated by recording sound levels and analyzing frequency components in an empty experimental chamber. There was no measurable noise or change in noise with the system active (Supplemental Information Fig. S15).**”

Additional Figures:

Figure S15. Noise characterization of sound chamber without animal occupation. (a) Amplitude of noise with magnetic field on at 10 W, (b) frequency analysis with magnetic field on at 10 W, (c) amplitude of noise with magnetic field off, (d) frequency analysis with magnetic field off. (e) Comparison of noise levels in chamber with magnetic field off and magnetic field on at 10 W.

Comment 6:

On the stimulation control and characterization. It would be necessary to clarify the exact light power delivered with the LEDs in this configuration. It would also be necessary to report if the device is suitable to deliver different light powers. This is a key feature in optogenetic experiments.

Our Response:

We thank the reviewer for the comments and agree that characterization of LED light intensity is essential to understanding the operational parameters of the device. We have added information on the stimulation intensity of the LED and have provided a graph (Fig. S9) showing the operational parameters compliant with the power available throughout the enclosure.

Modification to the manuscript:

Page 9 we added:

“The protocol in the top half set of graphs of Fig. 3b sets the device to operate at a frequency of 30 Hz, a duty cycle of 5%, and with left injectable probe on. The same device in the bottom half set of graphs is set to 10 Hz, a duty cycle of 15%, with the right probe on. All devices were set to operate at 10 mW/mm², a graph of operating intensities and required electrical power^{1,20} can be found in Supplemental Information Fig. S9.”

Additional Figures:

Figure S9. Graph of intensities and required average operational powers for each intensity (15% duty cycle).

Comment 7:

Regarding the temperature measurements, is nice to have a thermometer in the head of the animal and perhaps evaluate changes in the temperature along the light dark cycle. But perhaps it would be more useful to have a physiological quantification that its actually related to the optical stimulation. Is the thermistor sensitive enough to detect changes in the time scales of the optical stimulations? Was that measured? It would be helpful to have at least a section dedicated to the potential advantages of having these two features, the way it is presented seems unrelated.

Our Response:

We thank the reviewer for their comments and agree that thermal measurements during light and dark cycles represent an important capability of the technology presented here. The thermal recording was intended as a technological demonstrator to showcase that we have the ability to record and stimulate at the same time. Because characterization of temperature and effects of optogenetic stimulation and resulting changes in thermography are not well explored in songbirds, we have characterized the fidelity of the system by recording the circadian rhythm. Detailed investigations on physiological changes induced by optogenetic stimulation can be the subject of future investigations. To set the technological foundation, we have characterized the system temporal response of the thermal sensor to highlight fidelity of the device for future investigations. We also would like to note that we recently have demonstrated photometry capabilities to record GCAMP in rodents(R4). Given the technological improvements presented in this manuscript it is possible to combine this feature with optogenetic stimulation in future work.

References:

(R4) A. Burton, S. N. Obaid, A. Vázquez-Guardado, M. B. Schmit, T. Stuart, L. Cai, Z. Chen, I. Kandela, C. R. Haney, E. A. Waters, H. Cai, J. A. Rogers, L. Lu, P. Gutruf, Wireless, battery-free subdermally implantable photometry systems for chronic recording of neural dynamics. *Proc. Natl. Acad. Sci.*, 201920073 (2020).

Modification to the manuscript:

Page 11 we added:

“A calibration curve for corresponding ADC values and temperatures is provided in Fig. 4c over a dynamic range of 7.37 K indicating a resolution of 1.8 mK and accuracy of ± 0.097 K when comparing against a digital thermocouple thermometer (Proster). **The thermographic recording capabilities of the device can be used for a range of applications including chronic measurements of sleep-wake cycles in animal subjects as well as recording millisecond-level temperature changes as shown in Supplemental Information Fig. S13.**”

Additional Figures:

Figure S13. Graphs showing thermistor response to pulsed increase in temperature. (a) Temperature sensitivity at varying input powers of a heating coil placed in contact with the sensor and (b) temporal sensitivity of thermistor with heat applied at 6 Hz, 15% duty cycle, and 0.75 mW (red lines).

Comment 8:

My main concern regarding the results section and the paper in general, is the absence of a fundamental experiment demonstrating the possibility to modify neuronal activity. How do we know that the device works properly if we don't have any direct evidence attesting the change of neural activity? The main objective of optogenetic stimulation is to accurately manipulate neural activity, hence its fundamental to provide direct evidence that the new device is actually useful for that objective. For example, in figure 5, authors report a change in particular syllables of the song motif. However, we don't know if those changes are actually related to a change in neural activity or other factors, such a change in luminescence in the animal's skull making the animal distracted. Is this device compatible with electrophysiological quantifications?

Our Response:

We thank the reviewer for their comments and have clarified/expanded on the robust song results following optogenetic stimulation in Area X (see also Supplemental Information Figs. S16-20 /Table ST1) and have provided proof of expression of the opsins in the target area. The current configuration of the device does not include the capability to record neuronal firing but we hope to include that capability in a future design. Separately, wireless devices have been shown to effectively stimulate optogenetically in rodents(R5) and optogenetic stimulation in birds has been achieved recently with conventional fiber bound methods(R1-R3). Based on our extensive new data and this prior work of other groups, we are confident that the devices are eliciting optogenetic stimulation. This study is a proof of principal demonstration of the new technology; future mechanistic studies will investigate optogenetic stimulation of finch song nucleus Area X further, making extensive use of the devices presented here.

References:

- (R1) Xiao, G. Chattree, F. G. Oscos, M. Cao, M. J. Wanat, T. F. Roberts, A Basal Ganglia Circuit Sufficient to Guide Birdsong Learning. *Neuron*. **98**, 208-221.e5 (2018).
- (R2) W. Zhao, F. Garcia-Oscos, D. Dinh, T. F. Roberts, *Science* **366**, 83-89 (2019) doi:10.1126/science.aaw4226.
- (R3) T. F. Roberts, S. M. H. Gobes, M. Murugan, B. P. Ölveczky, R. Mooney, Motor circuits are required to encode a sensory model for imitative learning. *Nat. Neurosci.* **15**, 1454–1459 (2012).
- (R5) G. Shin, A. M. Gomez, R. Al-Hasani, Y. R. Jeong, J. Kim, Z. Xie, A. Banks, S. M. Lee, S. Y. Han, C. J. Yoo, J.-L. Lee, S. H. Lee, J. Kurniawan, J. Tureb, Z. Guo, J. Yoon, S.-I. Park, S. Y. Bang, Y. Nam, M. C. Walicki, V. K. Samineni, A. D. Mickle, K. Lee, S. Y. Heo, J. G. McCall, T. Pan, L. Wang, X. Feng, T. Kim, J. K. Kim, Y. Li, Y. Huang, R. W. Gereau, J. S. Ha, M. R. Bruchas, J. A. Rogers, Flexible Near-Field Wireless Optoelectronics as Subdermal Implants for Broad Applications in Optogenetics. *Neuron*. **93**, 509-521.e3 (2017).

Modification to the manuscript:

Page 14 we added:

“Proof of concept stimulation capabilities of the device are tested by targeting Area X, a song-dedicated basal ganglia brain nucleus in adult male zebra finches. We use viral delivery pathways established in a prior study by Xiao et al. using adeno-associated virus expressing human channel rhodopsin into the Ventral Tegmental Area (VTA) and taken up by dopaminergic neurons whose axons project to Area X. Prior studies show that continuous optogenetic stimulation of this VTA to Area X pathway over multiple days in tethered adult male zebra finches can alter their ability to pitch shift individual syllables within their songs in a learning task¹¹. Here, we use their same viral vector and targeting strategy (Supplemental Information Fig. S16) to stimulate Area X unilaterally over a single, short session (15-30 minutes) and elicit pitch shifts to demonstrate proof of principle. Stimulation parameters of 20 Hz and 15% duty cycle were used during the experimental sessions. Histological assessment confirms opsin expression and targeting of the probe (Supplemental Information Fig. S16).”

Citations added:

- (11) L. Xiao, G. Chattree, F. G. Oscos, M. Cao, M. J. Wanat, T. F. Roberts, A Basal Ganglia Circuit Sufficient to Guide Birdsong Learning. *Neuron*. **98**, 208-221.e5 (2018).

Page 15 we added:

“The basic unit of the bird’s song is a motif and is displayed as a spectrogram where individual syllables within the motif are identified based on their structural characteristics (Fig. 5c, f). Here, measurements of fundamental frequency (f_0 , pitch) are only made from syllables within the song that have a clear, uniform harmonic structure, as per established criteria in the field²³ (Supplemental Information Figs. S17, 18, 19). Each syllable is then scored across 25 consecutive song renditions to examine millisecond by millisecond changes in f_0 (see Methods). Song analysis reveals that across multiple subjects, a statistically significant shift (downward-Subjects A, B or upward-Subject C) in the f_0 is detected with stimulation (Fig. 5d-e, g-h). Specifically, stimulation-induced f_0 changes compared to pre-stimulation values are shown for Subject A (syllables B, C, D, F, G), Subject B (C, E, but not B), and Subject C (D, E) (Supplemental Information Table ST1/Supplemental Information Fig. S20).”

Citations added:

- (23) M. H. Kao, A. J. Doupe, M. S. Brainard, Contributions of an avian basal ganglia–forebrain circuit to real-time modulation of song. *Nature*. **433**, 638–643 (2005).

Additional Figures:

Figure S16. (a) Nissl staining in Area X-basal ganglia region. White arrows indicate spherical shape of Area X and the higher density of Nissl bodies within it. The μ -ILED probe track is shown with direction of light scatter (arrow). (b) Inset from A shown at higher magnification. Probe diameter $\sim 300\mu\text{m}$. (c) Red signal denotes staining for tyrosine hydroxylase, the enzyme for dopamine biosynthesis, in coronal brain section three to four weeks after virus injection. (d) Green signal denotes virally-driven Green Fluorescent Protein (GFP) expression. (e) Merged image of c-d. (f) Inset from c at higher magnification. (g) Inset from d at higher magnification. (h) Inset from e at higher magnification. Arrows

indicate some examples of GFP signal in dopaminergic neurons. Coronal Section Orientation: D-dorsal, V-ventral, M-medial, L-lateral. See Methods: Tissue Histology.

Figure S17. (a) Spectrogram of birdsong motif from finch Subject A and individual harmonic syllables analyzed with arrow denoting where f_0 is measured for (b) B pre-stimulation, (c) B during stimulation, (d) C pre-stimulation, (e) C during stimulation; only the harmonic portion of this syllable was analyzed, (f) D pre-stimulation, (g) D during

stimulation, (h) F pre-stimulation, (i) F during stimulation, (j) G pre-stimulation, (k) G during stimulation.

Figure S18. (a) Spectrogram of birdsong motif from finch Subject B and individual syllables harmonic portions analyzed with arrow denoting where f_0 is measured for (b) B pre-stimulation, (c) B during stimulation, (d) C pre-stimulation, (e) C during stimulation, (f) E pre-stimulation, (g) during stimulation

Figure S19. (a) Spectrogram of birdsong motif from finch Subject C and individual syllables with arrow denoting where f_o is measured for (b) D pre-stimulation, (c) D during stimulation, with two f_o s denoted, (d) E pre-stimulation, (e) E during stimulation; only harmonic portion of E was analyzed.

Figure S20. Fundamental frequency (f_0) scores of pre-stimulation (black line) and post stimulation (blue line) over 25 copies for (a) subject A and syllable B, (b) syllable C, (c) syllable D, (d) syllable F, (e) syllable G, (f) subject B and syllable C, (g) syllable E, (h) subject C and syllable D f_{o2} , (i) syllable E. Pre-stimulation versus stimulation comparisons for each plot were significant at $p < 0.05$, Wilcoxon signed rank; see Supplemental Table ST1 for p-values and Methods for statistical analyses.

Additional Tables:

Subject	Syllable	Pre-Stimulation		Stimulation	
		Mean	Standard Deviation	Mean	Standard Deviation
A	B, p<0.000	635.497	4.927	623.994	8.141
	C, p<0.001	566.439	5.262	559.031	6.048
	D, p<0.000	543.542	7.433	532.592	6.101
	F, p<0.000	710.094	5.881	700.524	4.355
	G, p<0.017	560.79	8.618	555.415	7.694
B	B, p<0.353	608.712	5.562	606.544	4.997
	C, p<0.001	555.849	8.051	548.342	5.974
	E, p<0.007	682.138	7.54	676.603	4.373
C	D, fo1, p<0.26	1080.28	20.776	1074.82	18.049
	D, fo2, p<0.025	1241.64	23.49	1255.08	19.303
	E, p<0.001	571.269	6.385	578.061	5.285

Table ST1. Table documenting mean pitch score and standard deviation values over 25 copies of either pre-stimulation or stimulation recordings.

Comment 9:

It's not clear how the statistical analysis was performed, for instance, the syllables showing statistical differences were specifically selected for the comparison or on the contrary, all syllables were analyzed, and the ones reported were the only ones showing statistical differences? In any case, this should be explicitly clarified in results and methods sections, since this is the only evidence attesting for the functionality of the device. It would also be necessary to compare syllables before, during and after stimulations and not only before and during stimulations.

Our Response:

We have now clarified the statistical analysis and selection of syllable type in the text as well as provided all pitch data across bird subjects in the Results and Supplemental Data sections. In brief, fundamental frequency (pitch) analyses is only performed on those syllables that have a clear harmonic structure within the bird's song. Each bird has multiple syllables that are either pure harmonics or contain harmonic notes, both subtypes were analyzed and are indicated in Fig. 5c, f and Supplemental Information Figs. S17, 18, 19. Pitch scores are obtained from 25 consecutive copies of each syllable within the bird's song; 25 provides sufficient power to detect experimental differences based on our prior studies (R4, R5). After assessing the data distribution for normality, non-parametric tests were performed (Wilcoxon), comparing across pre-stimulation and stimulation time points within the same finch. Based on our current dataset, evaluation of the after-stimulation period would require a more in-depth biological study and a much larger sample size to determine the time course for restoration of song following stimulation, which is out of the scope of this manuscript. The reason why we cannot simply obtain this data is because of motivational behavior. Specifically, it is very hard to get long sequences of bird song consistently to do extensive analysis because the song is vocalized voluntarily by the subject, and there is no means to stimulate the song on demand. Xiao et al. showed that continuous daily optogenetic stimulation of Area X using this same virus results in significant pitch shifts over many days during a pitch learning task(R6). In our future directions in this research, we plan to evaluate these biological mechanisms further using our specific paradigm.

References:

- (R4) J. E. Miller, G. W. Hafzalla, Z. D. Burkett, C. M. Fox, S. A. White, Reduced vocal variability in a zebra finch model of dopamine depletion: implications for Parkinson disease. *Physiol. Rep.* **3**, e12599 (2015).
- (R5) J. E. Miller, A. T. Hilliard, S. A. White, Song practice promotes acute vocal variability at a key stage of sensorimotor learning. *PLoS One.* **5**, e8592–e8592 (2010).
- (R6) L. Xiao, G. Chattree, F. G. Oscos, M. Cao, M. J. Wanat, T. F. Roberts, A Basal Ganglia Circuit Sufficient to Guide Birdsong Learning. *Neuron.* **98**, 208-221.e5 (2018).

Modification to the manuscript:

Please refer to Comment 8, additional text page 15

Page 25 we added:

“Zebra finch song consists of a sequence of repeated syllables that comprise a motif (Fig. 5c, f **Supplemental Information Figs. S17, 18, 19**). Using SAP, and a semi-automated clustering program (VOICE)^{29,30}, wav files for 25 consecutive individual syllables with a harmonic stack component were identified (**Supplemental Information Fig. S17: Subject A-Syllables B, C, D, F, G; Supplemental Information Fig. S18: Subject B-Syllables B, C, E; Supplemental Information Fig. S19: Subject C-Syllables D, E**). Prior power analyses determined that 25 consecutive syllable copies within a bird are sufficient to detect meaningful differences based on experimental condition³¹. The wav files for 25 harmonic syllables were then run in Matlab version R2014a using the SAP SAT Tools **and PRAAT** to obtain measurements of fundamental frequency, f_0 (pitch) as per prior work^{23,32}. Individual f_0 values were plotted from 25 consecutive copies just prior to optical stimulation, ‘Pre-Stimulation’ and compared to 25 copies of the same syllable during the ‘Stimulation’ period at 20Hz/15P (**Supplemental Information Fig. 20**). **Statistical Analyses: Because the syllable data did not fit a normal distribution, we used the non-parametric Wilcoxon signed-rank test for paired data in which the scores for the same syllable were compared between pre-stimulation and stimulation periods.** Significance was set at $p < 0.05$ (IBM SPSS Statistics for Windows version 26, Armonk, NY), **and p-values are reported in Supplemental Information Table ST1.**”

Citations added:

- (29) O. Tchernichovski, F. Nottebohm, C. E. Ho, B. Pesaran, P. P. Mitra, A procedure for an automated measurement of song similarity. *Anim. Behav.* **59**, 1167–1176 (2000).
- (32) A. Badwal, M. Borgstrom, R. A. Samlan, J. E. Miller, Middle age, a key time point for changes in birdsong and human voice. *Behav. Neurosci.* **134**, 208–221 (2020).

Additional Figures:

Please refer to Comment 8, additional Figures S17, S18, S19, S20

Additional Tables:

Please refer to Comment 8, additional Table ST1

Comment 10:

It's also noticeable the absence of histological confirmation of infection sites and the potential tissue damage produce by the insertion of the optical devices. Since there's no direct evidence on neural control, this data becomes fundamental to understand the behavioral effects.

Our Response:

We thank the reviewer for their comments and agree that histological data is important for verifying the confirmed infection sites. We have added histological data and evidence of opsin expression.

Modification to the manuscript:

Please refer to Comment 8, additional text page 14

Page 26 we added:*“Tissue Histology*

Finches that received the AAV injection into VTA and optogenetic device implant into Area X were humanely euthanized with an overdose of isoflurane inhalant and then transcardially perfused with warmed saline followed by chilled 4% paraformaldehyde in Dulbecco's Phosphate Buffer Saline. Fixed brains were cryoprotected in 20% sucrose overnight and then coronally sectioned at 30µm on a Microm cryostat. Tissue was processed for fluorescent immunohistochemistry, using a procedure similar to Miller et al. 2015³³: Hydrophobic borders were drawn on the slides using a pap pen (ImmEdge, Vector Labs) followed by 3 X 5 minute washes in 1X TBS with 0.3% Triton X (Tx). To block non-specific antibody binding, the tissue was then incubated for one hour (hr) at room temperature with 5% goat serum (Sigma-Aldrich #G-9023) in TBS/0.3% Tx followed by 3 x 5 minute washes in 1% goat serum in TBS/0.3% Tx. Primary antibodies were incubated in a solution of 1% goat serum in TBS/0.3% Tx overnight at 4°C. For the VTA/SNc region, a primary rabbit polyclonal antibody was applied (1:500 of Tyrosine Hydroxylase-TH, Millipore Sigma #AB152) to mark dopaminergic cell bodies with a primary mouse monoclonal antibody to detect virus expression via Green Fluorescence Protein (1:100, ThermoFisher 3E6, #A11120). A “no primary antibody” control was performed during initial testing. The following day, sections were washed 5 x 5 minutes in 1x TBS/0.3% Tx and incubated for three hrs at room temperature in fluorescently conjugated secondary antibodies in 1% goat serum with 1x TBS/0.3% Tx (ThermoFisher 1:1000, goat anti-rabbit 647 #A-21245 for TH and goat anti-mouse 568 #A11031 for GFP). After secondary incubation, sections were washed 3 x 10 minutes in TBS followed by 2 x 5 minute washes in filtered TBS. Slides were then cover-slipped in Pro-Long Anti-Fade Gold mounting medium (Invitrogen, #P36930) and imaged on a Leica DMI6000B with a DFC450 color CCD camera (Leica Microsystems, Buffalo Grove, IL) using the Leica LAS-X version 3.3 software.” To assess whether the optogenetic probe induced damage in Area X, tissue

sections were processed through a Nissl staining procedure (1% thionin, Fishersci #50520580), cleared in xylene (Fishersci X5-4), mounted in DPX (Sigma-Aldrich #6522), and visualized using a Nikon Eclipse E800 bright field stereoscope connected to an Olympus color CCD DP73 camera and CellSens software.”

Citations added:

- (33) J. E. Miller, G. W. Hafzalla, Z. D. Burkett, C. M. Fox, S. A. White, Reduced vocal variability in a zebra finch model of dopamine depletion: implications for Parkinson disease. *Physiol. Rep.* **3**, e12599 (2015).

Page 28 we added:

“The authors thank T.F. Roberts of the University of Texas Southwestern Medical Center in Dallas for the viral vector, Kent Clemence for cage construction, and University of Arizona Animal Care. We also thank Mr. Douglas W. Cromey for training and use of the Leica DMI6000 through the University of Arizona, Imaging Cores - Life Sciences North under a Core Facilities Pilot Program awarded to J. E.M. J.E.M. also acknowledges start up funds from the Depts. of Neuroscience and Speech, Language and Hearing Sciences.”

Additional Figures:

Please refer to Comment 8, additional Figure S16

Comment 11:

The discussion is not actually a discussion but a summary of the results section, therefore it must be rewritten to include the limitations of the current design, the possibilities for improving it or combining it with other techniques for neural interrogations.

Our Response:

We thank the reviewer for their comments and agree that the discussion section needs to be modified to include limitations of the current design as well as possibilities for improving and combining it with other techniques. We included discussion on limitations and synergy with current techniques.

Modification to the manuscript:**Page 17 we added:**

“The combination of these advances provides a toolbox towards the device design for investigation platforms in freely **flying** subjects that substantially expands the operational conditions of wireless, battery free and subdermally implantable modulation and recording tools for the central nervous system.

The current device embodiment features thermal recording capabilities highlighting the ability to include a variety of sensors. Examples for possible future recording capabilities include photometric probes which have been demonstrated in rodent subjects² and electrophysiological recording to capture non cell specific firing activity. The advances presented in this work provide foundational design approaches to enable tools for studies in songbirds for complex social interactions and chronic changes in song and behavior.”

Comment 12:

In figure 5, it would be useful to include standard deviation indicators in subpanels d-e and g-h. To have a visual comparison of the phenomena, it would also be nice to have spectrograms extracted from stimulated periods.

Our Response:

We thank the reviewer for their comments. To clarify, Fig. 5 data shows individual values plotted for each syllable type over 25 song renditions to illustrate the shift in pitch that occurs during stimulation compared to pre-stimulation of the same syllable. Birds naturally show variation in pitch as they sing so we highlight this variation (standard deviation-SD) from the mean scores in Supplemental Information Table ST1. We opt not to add the SD values in the figures themselves because of additional clutter that detracts from the overall view of the data. We also include spectrograms of individual syllable examples from the stimulation period for Subjects A-C (Supplemental Information Figs. S17, 18, 19) to show different syllable types and how the measurement is made. Pitch changes are subtle and cannot easily be detected by eye.

Modification to the manuscript:

Please refer to Comment 9, additional text page 25

Additional Figures:

Please refer to Comment 8, additional Figures S17, S18, S19

Additional Tables:

Please refer to Comment 8, additional Table ST1

Comment 13:

Would it be possible to know the places where the animals were stimulated? In figure 5b, it's remarkable that the vast majority of the time, the animals remained in the upper part of the cage, but the distance traveled is in the range of kilometers, does that mean that the animals moved a lot on that small upper space? Presented like that, it's not very clear how this device is ideal for the so-called 3D moving species.

Our Response:

We thank the reviewer for their comments and agree that Fig. 5b shows that the animal spends the majority of its time perched in the upper half of the cage yet the distance traveled is in the range of kilometers. The graph needs an activity cutoff to clearly show the movement of the subject. We have included an alternative way to plot subject activity with a tracemap in the Supplemental Fig. S4 to highlight the movement pattern of the birds. It can be seen that the subject frequently travels from the top perch to the bottom where their food is located. Because that movement is quick (flying) it does not show up in the color graph in the main figure.

Figure S4. Trace tracking plots for the (a) side and (b) top view of the experimental arena created with SimBA software at a 99% confidence rate cutoff.

Modification to the manuscript:

Page 6 we added:

“The behavioral pattern indicates that most of the time is spent in the upper and lower half of the arena with close proximity to the arena walls. **This behavioral pattern is maintained throughout the day as the bird frequently flies to the floor of the arena to get food and then returns to the perch located near the top of the arena.**”

Response to comments of Referee #3

Reviewer #3 (remarks to the author):

General Comments:

This paper reports the development of a wireless, battery free and multimodal platform that enables optogenetic stimulation and physiological recording in a miniaturized form factor for use in songbirds. Additionally, the design approaches used by authors were able to expand the use of wireless subdermally implantable neuromodulation and sensing tools to species previously excluded from in vivo real time experiments.

The paper is interesting, quite well structured and written, with good results and well discussed. Therefore, I recommend this paper for publication in the Nature Communications edition.

Nevertheless, the paper would benefit from a few minor corrections to achieve better clarity to the reader:

Our Response:

We thank the reviewer for their positive assessment of impact and appreciate the insightful comments that we have addressed with additional experiments.

Modification to the manuscript: none

Comment 1:

Page 18: "Videos were recorded with 2 cameras (Anivia 1080p HD Webcam W8, 1920*1080, 30 FPS) mounted above and in front of the arena for a bird with implant (n=1) and a bird without implant (n=1)." Why only n=1 was used? Is this statistically significant?

Our Response:

We thank the reviewer for their comments and agree that the low n is indeed limiting insight into general behavior of the subjects. To address this we have increased the number of subjects and trials to come to a statistically significant conclusion. Specifically, n = 3 for bird with implant and n = 2 for bird without implant.

Each of the presented datapoints represents a 14 hour recording and tracing of the birds activity resulting in a sum of 70 hours of analyzed behaviour.

Modification to the manuscript:

Page 22 we added:

"Videos were recorded with 2 cameras (Anivia 1080p HD Webcam W8, 1920*1080, 30 FPS) mounted above and in front of the arena for a bird with thermography implant (n=3) and a bird without implant (n=2)."

Additional Figures:

Figure 5b. Heat maps showing animal model position over a 14-hour period imposed on a 3D rendering of the experimental arena for control and implanted subject (left). Measurement of distance traveled for a control and implanted subject (right).

Comment 2:

The same can be observed in page 19: “Implanted birds (n=2) were acclimated in the testing arena prior to experiments”. Why was used only n=2?

Our Response:

We thank the reviewer for their comments and agree that a higher n is required to increase the statistical significance of the results. It is important to note that each subject produces many rendition of songs, and we analyzed a total of 25 song renditions before and during optogenetic stimulation across multiple harmonic type syllables. This results in a very large dataset that documents the effects of optogenetic stimulation. To add to this already large dataset, we have completed analysis of another subject and have included the results in the manuscript. With the new subject significant pitch shifts can also be observed, further confirming optogenetic stimulation capability of our platform.

Modification to the manuscript:**Page 23 we added:**

“Implanted birds (n=3) were acclimated in the testing arena prior to experiments.”

Additional Figures:

Figure S20. Fundamental frequency (f_0) scores of pre-stimulation (black line) and post stimulation (blue line) over 25 copies for (a) subject A and syllable B, (b) syllable C, (c) syllable D, (d) syllable F, (e) syllable G, (f) subject B and syllable C, (g) syllable E, (h) subject C and syllable D f_{02} , (i) syllable E. Pre-stimulation versus stimulation comparisons for each plot were significant at $p < 0.05$, Wilcoxon signed rank; see Supplemental Table ST1 for p-values and Methods for statistical analyses.

References

- (1) Gutruf, P.; Krishnamurthi, V.; Vázquez-Guardado, A.; Xie, Z.; Banks, A.; Su, C.-J.; Xu, Y.; Haney, C. R.; Waters, E. A.; Kandela, I.; Krishnan, S. R.; Ray, T.; Leshock, J. P.; Huang, Y.; Chanda, D.; Rogers, J. A. Fully Implantable Optoelectronic Systems for Battery-Free, Multimodal Operation in Neuroscience Research. *Nat. Electron.* **2018**, *1* (12), 652–660. <https://doi.org/10.1038/s41928-018-0175-0>.
- (2) Burton, A.; Obaid, S. N.; Vázquez-Guardado, A.; Schmit, M. B.; Stuart, T.; Cai, L.; Chen, Z.; Kandela, I.; Haney, C. R.; Waters, E. A.; Cai, H.; Rogers, J. A.; Lu, L.; Gutruf, P. Wireless, Battery-Free Subdermally Implantable Photometry Systems for Chronic Recording of Neural Dynamics. *Proc. Natl. Acad. Sci.* **2020**, 201920073. <https://doi.org/10.1073/pnas.1920073117>.
- (3) Gutruf, P.; Yin, R. T.; Lee, K. B.; Ausra, J.; Brennan, J. A.; Qiao, Y.; Xie, Z.; Peralta, R.; Talarico, O.; Murillo, A.; Chen, S. W.; Leshock, J. P.; Haney, C. R.; Waters, E. A.; Zhang, C.; Luan, H.; Huang, Y.; Trachiotis, G.; Efimov, I. R.; Rogers, J. A. Wireless, Battery-Free, Fully Implantable Multimodal and Multisite Pacemakers for Applications in Small Animal Models. *Nat. Commun.* **2019**, *10* (1), 5742. <https://doi.org/10.1038/s41467-019-13637-w>.
- (4) Zhang, Y.; Castro, D. C.; Han, Y.; Wu, Y.; Guo, H.; Weng, Z.; Xue, Y.; Ausra, J.; Wang, X.; Li, R. Battery-Free, Lightweight, Injectable Microsystem for in Vivo Wireless Pharmacology and Optogenetics. *Proc. Natl. Acad. Sci.* **2019**, 201909850.
- (5) Zhang, Y.; Mickle, A. D.; Gutruf, P.; McIlvried, L. A.; Guo, H.; Wu, Y.; Golden, J.

- P.; Xue, Y.; Grajales-Reyes, J. G.; Wang, X.; Krishnan, S.; Xie, Y.; Peng, D.; Su, C.-J.; Zhang, F.; Reeder, J. T.; Vogt, S. K.; Huang, Y.; Rogers, J. A.; Gereau, R. W. Battery-Free, Fully Implantable Optofluidic Cuff System for Wireless Optogenetic and Pharmacological Neuromodulation of Peripheral Nerves. *Sci. Adv.* **2019**, *5* (7), eaaw5296. <https://doi.org/10.1126/sciadv.aaw5296>.
- (6) Lu, L.; Gutruf, P.; Xia, L.; Bhatti, D. L.; Wang, X.; Vazquez-Guardado, A.; Ning, X.; Shen, X.; Sang, T.; Ma, R. Wireless Optoelectronic Photometers for Monitoring Neuronal Dynamics in the Deep Brain. *Proc. Natl. Acad. Sci.* **2018**, *115* (7), E1374–E1383.
- (7) Shin, G.; Gomez, A. M.; Al-Hasani, R.; Jeong, Y. R.; Kim, J.; Xie, Z.; Banks, A.; Lee, S. M.; Han, S. Y.; Yoo, C. J.; Lee, J.-L.; Lee, S. H.; Kurniawan, J.; Tureb, J.; Guo, Z.; Yoon, J.; Park, S.-I.; Bang, S. Y.; Nam, Y.; Walicki, M. C.; Samineni, V. K.; Mickle, A. D.; Lee, K.; Heo, S. Y.; McCall, J. G.; Pan, T.; Wang, L.; Feng, X.; Kim, T.; Kim, J. K.; Li, Y.; Huang, Y.; Gereau, R. W.; Ha, J. S.; Bruchas, M. R.; Rogers, J. A. Flexible Near-Field Wireless Optoelectronics as Subdermal Implants for Broad Applications in Optogenetics. *Neuron* **2017**, *93* (3), 509-521.e3. <https://doi.org/https://doi.org/10.1016/j.neuron.2016.12.031>.
- (8) Hisey, E.; Kearney, M. G.; Mooney, R. A Common Neural Circuit Mechanism for Internally Guided and Externally Reinforced Forms of Motor Learning. *Nat. Neurosci.* **2018**, *21* (4), 589–597. <https://doi.org/10.1038/s41593-018-0092-6>.
- (9) Sakata, J.; Woolley, S.; Fay, R.; Popper, A. *The Neuroethology of Birdsong*; 2020. <https://doi.org/10.1007/978-3-030-34683-6>.
- (10) Kearney, M. G.; Warren, T. L.; Hisey, E.; Qi, J.; Mooney, R. Discrete Evaluative

- and Premotor Circuits Enable Vocal Learning in Songbirds. *Neuron* **2019**, *104* (3), 559-575.e6. <https://doi.org/https://doi.org/10.1016/j.neuron.2019.07.025>.
- (11) Xiao, L.; Chattree, G.; Ocos, F. G.; Cao, M.; Wanat, M. J.; Roberts, T. F. A Basal Ganglia Circuit Sufficient to Guide Birdsong Learning. *Neuron* **2018**, *98* (1), 208-221.e5. <https://doi.org/https://doi.org/10.1016/j.neuron.2018.02.020>.
- (12) Kale, R. P.; Kouzani, A. Z.; Walder, K.; Berk, M.; Tye, S. J. Evolution of Optogenetic Microdevices. *Neurophotonics* **2015**, *2* (3), 31206. <https://doi.org/10.1117/1.NPh.2.3.031206>.
- (13) Miyamoto, D.; Murayama, M. The Fiber-Optic Imaging and Manipulation of Neural Activity during Animal Behavior. *Neurosci. Res.* **2016**, *103*, 1–9. <https://doi.org/https://doi.org/10.1016/j.neures.2015.09.004>.
- (14) Land, B. B.; Brayton, C. E.; Furman, K.; LaPalombara, Z.; Dileone, R. Optogenetic Inhibition of Neurons by Internal Light Production. *Front. Behav. Neurosci.* **2014**, *8*.
- (15) Nager, R. G.; Law, G. The Zebra Finch. *The UFAW Handbook on the Care and Management of Laboratory and Other Research Animals*. March 26, 2010, pp 674–685. <https://doi.org/doi:10.1002/9781444318777.ch43>.
- (16) Kawakami, M.; Yamamura, K. Cranial Bone Morphometric Study among Mouse Strains. *BMC Evol. Biol.* **2008**, *8* (1), 73. <https://doi.org/10.1186/1471-2148-8-73>.
- (17) Samineni, V. K.; Yoon, J.; Crawford, K. E.; Jeong, Y. R.; McKenzie, K. C.; Shin, G.; Xie, Z.; Sundaram, S. S.; Li, Y.; Yang, M. Y. Fully Implantable, Battery-Free Wireless Optoelectronic Devices for Spinal Optogenetics. *Pain* **2017**, *158* (11), 2108.

- (18) Mathis, A.; Mamidanna, P.; Cury, K. M.; Abe, T.; Murthy, V. N.; Mathis, M. W.; Bethge, M. DeepLabCut: Markerless Pose Estimation of User-Defined Body Parts with Deep Learning. *Nat. Neurosci.* **2018**, *21* (9), 1281–1289. <https://doi.org/10.1038/s41593-018-0209-y>.
- (19) Schormans, M.; Valente, V.; Demosthenous, A. Practical Inductive Link Design for Biomedical Wireless Power Transfer: A Tutorial. *IEEE Trans. Biomed. Circuits Syst.* **2018**, *12* (5), 1112–1130. <https://doi.org/10.1109/TBCAS.2018.2846020>.
- (20) Zhang, Y.; Castro, D. C.; Han, Y.; Wu, Y.; Guo, H.; Weng, Z.; Xue, Y.; Ausra, J.; Wang, X.; Li, R.; Wu, G.; Vázquez-Guardado, A.; Xie, Y.; Xie, Z.; Ostojich, D.; Peng, D.; Sun, R.; Wang, B.; Yu, Y.; Leshock, J. P.; Qu, S.; Su, C.-J.; Shen, W.; Hang, T.; Banks, A.; Huang, Y.; Radulovic, J.; Gutruf, P.; Bruchas, M. R.; Rogers, J. A. Battery-Free, Lightweight, Injectable Microsystem for in Vivo Wireless Pharmacology and Optogenetics. *Proc. Natl. Acad. Sci.* **2019**, *116* (43), 21427 LP – 21437. <https://doi.org/10.1073/pnas.1909850116>.
- (21) Aronov, D.; Fee, M. S. Natural Changes in Brain Temperature Underlie Variations in Song Tempo during a Mating Behavior. *PLoS One* **2012**, *7* (10), e47856.
- (22) Zhang, H.; Gutruf, P.; Meacham, K.; Montana, M. C.; Zhao, X.; Chiarelli, A. M.; Vázquez-Guardado, A.; Norris, A.; Lu, L.; Guo, Q. Wireless, Battery-Free Optoelectronic Systems as Subdermal Implants for Local Tissue Oximetry. *Sci. Adv.* **2019**, *5* (3), eaaw0873.
- (23) Kao, M. H.; Doupe, A. J.; Brainard, M. S. Contributions of an Avian Basal Ganglia–Forebrain Circuit to Real-Time Modulation of Song. *Nature* **2005**, *433* (7026), 638–643. <https://doi.org/10.1038/nature03127>.

- (24) He, W.; Goudeau, P.; Le Bourhis, E.; Renault, P.-O.; Dupré, J. C.; Doumalin, P.; Wang, S. Study on Young's Modulus of Thin Films on Kapton by Microtensile Testing Combined with Dual DIC System. *Surf. Coatings Technol.* **2016**, *308*, 273–279. <https://doi.org/https://doi.org/10.1016/j.surfcoat.2016.07.114>.
- (25) Ledbetter, H. M.; Naimon, E. R. Elastic Properties of Metals and Alloys. II. Copper. *J. Phys. Chem. Ref. Data* **1974**, *3* (4), 897–935. <https://doi.org/10.1063/1.3253150>.
- (26) Rizzi, F.; Qualtieri, A.; Chambers, L. D.; Megill, W. M.; De Vittorio, M. Parylene Conformal Coating Encapsulation as a Method for Advanced Tuning of Mechanical Properties of an Artificial Hair Cell. *Soft Matter* **2013**, *9* (9), 2584–2588. <https://doi.org/10.1039/C2SM27566J>.
- (27) Nilsson, S. R. O.; Goodwin, N. L.; Choong, J. J.; Hwang, S.; Wright, H. R.; Norville, Z. C.; Tong, X.; Lin, D.; Bentzley, B. S.; Eshel, N.; McLaughlin, R. J.; Golden, S. A. Simple Behavioral Analysis (SimBA) – an Open Source Toolkit for Computer Classification of Complex Social Behaviors in Experimental Animals. *bioRxiv* **2020**, 2020.04.19.049452. <https://doi.org/10.1101/2020.04.19.049452>.
- (28) Lukas. Heatscatter Plot for Variable X and Y. MATLAB Central File Exchange 2020.
- (29) Tchernichovski, O.; Nottebohm, F.; Ho, C. E.; Pesaran, B.; Mitra, P. P. A Procedure for an Automated Measurement of Song Similarity. *Anim. Behav.* **2000**, *59* (6), 1167–1176. <https://doi.org/https://doi.org/10.1006/anbe.1999.1416>.
- (30) Burkett, Z. D.; Day, N. F.; Peñagarikano, O.; Geschwind, D. H.; White, S. A. VOICE: A Semi-Automated Pipeline for Standardizing Vocal Analysis across

- Models. *Sci. Rep.* **2015**, 5, 10237. <https://doi.org/10.1038/srep10237>.
- (31) Miller, J. E.; Hilliard, A. T.; White, S. A. Song Practice Promotes Acute Vocal Variability at a Key Stage of Sensorimotor Learning. *PLoS One* **2010**, 5 (1), e8592–e8592. <https://doi.org/10.1371/journal.pone.0008592>.
- (32) Badwal, A.; Borgstrom, M.; Samlan, R. A.; Miller, J. E. Middle Age, a Key Time Point for Changes in Birdsong and Human Voice. *Behav. Neurosci.* **2020**, 134 (3), 208–221. <https://doi.org/10.1037/bne0000363>.
- (33) Miller, J. E.; Hafzalla, G. W.; Burkett, Z. D.; Fox, C. M.; White, S. A. Reduced Vocal Variability in a Zebra Finch Model of Dopamine Depletion: Implications for Parkinson Disease. *Physiol. Rep.* **2015**, 3 (11), e12599. <https://doi.org/https://doi.org/10.14814/phy2.12599>.
- (R1) Xiao, G. Chattree, F. G. Oscos, M. Cao, M. J. Wanat, T. F. Roberts, A Basal Ganglia Circuit Sufficient to Guide Birdsong Learning. *Neuron*. **98**, 208-221.e5 (2018).
- (R2) W. Zhao, F. Garcia-Oscos, D. Dinh, T. F. Roberts, *Science (80-.)*., in press, doi:10.1126/science.aaw4226.
- (R3) T. F. Roberts, S. M. H. Gobes, M. Murugan, B. P. Ölveczky, R. Mooney, Motor circuits are required to encode a sensory model for imitative learning. *Nat. Neurosci.* **15**, 1454–1459 (2012).
- (R4) A. Burton, S. N. Obaid, A. Vázquez-Guardado, M. B. Schmit, T. Stuart, L. Cai, Z. Chen, I. Kandela, C. R. Haney, E. A. Waters, H. Cai, J. A. Rogers, L. Lu, P. Gutruf, Wireless, battery-free subdermally implantable photometry systems for chronic recording of neural dynamics. *Proc. Natl. Acad. Sci.*, 201920073 (2020).
- (R5) G. Shin, A. M. Gomez, R. Al-Hasani, Y. R. Jeong, J. Kim, Z. Xie, A. Banks, S. M.

Lee, S. Y. Han, C. J. Yoo, J.-L. Lee, S. H. Lee, J. Kurniawan, J. Tureb, Z. Guo, J. Yoon, S.-I. Park, S. Y. Bang, Y. Nam, M. C. Walicki, V. K. Samineni, A. D. Mickle, K. Lee, S. Y. Heo, J. G. McCall, T. Pan, L. Wang, X. Feng, T. Kim, J. K. Kim, Y. Li, Y. Huang, R. W. Gereau, J. S. Ha, M. R. Bruchas, J. A. Rogers, Flexible Near-Field Wireless Optoelectronics as Subdermal Implants for Broad Applications in Optogenetics. *Neuron*. **93**, 509-521.e3 (2017).

(R6) L. Xiao, G. Chattree, F. G. Ocos, M. Cao, M. J. Wanat, T. F. Roberts, A Basal Ganglia Circuit Sufficient to Guide Birdsong Learning. *Neuron*. **98**, 208-221.e5 (2018).

Reviewers' Comments:

Reviewer #1:

None

Reviewer #2:

Remarks to the Author:

The authors carefully addressed all my comments, I recognize the effort to clarify my concerns, particularly regarding my comment 8. The MS is much clearer, I have no further comments.

Reviewer #3:

None

Response to comments of Referee #2

Reviewer #2 (remarks to the author):

General Comments:

The authors carefully addressed all my comments, I recognize the effort to clarify my concerns, particularly regarding my comment 8. The MS is much clearer, I have no further comments.

Our Response:

We thank the reviewer for these comments and acknowledge that addressing the reviewer's concerns adds clarity to the manuscript.